# SMPE: A Framework for Multi-Dimensional Permutation Equivariance

## Abstract

Permutation equivariance (PE) is an important inductive prior for addressing tasks such as point cloud segmentation, where permuting objects in the input set maintains the output features of each object. However, the state-of-the-art PE methods mainly focus on one-dimensional cases, which cannot meet the requirements of multi-dimensional tasks such as auction design, pseudo inverse computation, and multiuser resource allocation in wireless networks. It is evidenced that the direct incorporation of high-dimensional equivariance in network design necessitates tensor operations and complicated parameter sharing patterns (Hartford et al., 2018), which contributes to its limited exploration. In this paper, we propose a novel serial multi-dimensional permutation equivariance (**SMPE**) framework to address these challenges. By serially composing multiple one-dimensional equivariant layers and incorporating dense connections for feature reuse, the proposed SMPE framework enables cross-dimensional interactions among objects while maintaining multi-dimensional equivariance. Additionally, we extend the SMPE framework to scenarios of permutation invariance as well as the hybrid equivariance and invariance through pooling operations. We use an extensive set of experiments to evaluate the framework on contextual auction design, pseudo inverse computation, and multiuser wireless communication tasks. It is observed that the SMPE framework not only maintains excellent equivariant properties to support variable set sizes but also outperforms the state-of-the-art models. For example, SMPE could gain as high as 10.6% and 7.1% improvements over the state-of-the-art methods in two typical multiuser resource allocation scenarios.

## 1 Introduction

Permutation equivariance (or equivariant for short) usually arises in tasks modeling interactions across a set of objects, meaning that the output features of individual objects remain unchanged when the order of objects is permuted. Zaheer et al. (2017). For instance, in tasks related to point clouds (Qi et al., 2017; Li & Lee, 2019), the point cloud can be regarded as an unordered set of points since input features contain the coordinate information of point clouds, implying that shuffling the order of points will not change the output point features. In tasks related to graphs (Wu et al., 2020; Zhou et al., 2020), permuting the input order of nodes does not affect the output features of the corresponding nodes since the topology of the graph remains unchanged. Designing networks with permutation equivariance to address related tasks is an important issue due to the following two significant advantages. The first is **parameter sharing** introduced by the equivariance (Ravanbakhsh et al., 2017; Ravanbakhsh, 2020). This results in the parameters amount of equivariant networks being independent of the number of objects. Such property leads to a significant reduction in computational complexity and presents an advantage when dealing with sets containing a large number of objects Hartford et al. (2018). The second is the ability to **handle sets of different sizes** (Zaheer et al., 2017). Since there is no dependency between parameters and the number of objects, the trained equivariant network can operate with different set sizes (Keriven & Peyré, 2019).

When considering unordered objects across multiple sets, permutation equivariance is extended to higher dimensions. For instance, tasks related to tabular data often involve interactions between two sets of objects Duan et al. (2022); Rahme et al. (2021), which can be modeled as a 2D equivariant mapping Maron et al. (2020). High-dimensional equivariant function $f$ is illustrated in figure 1. In

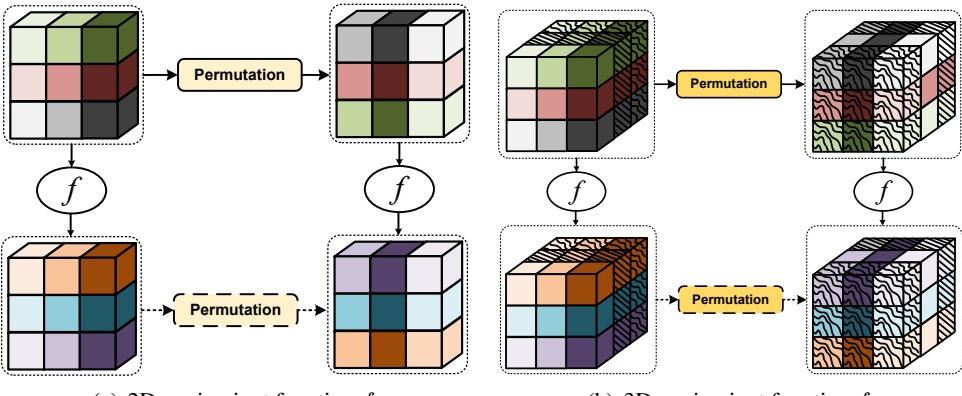

(a) 2D equivariant function $f$.      (b) 3D equivariant function $f$.

Figure 1: Multi-dimensional equivariant mapping $f$: The order of the output items will be permuted accordingly when we permute the input items across multiple dimensions. The permutations of rows and columns in Figure (a) are $\{1, 2, 3\} \rightarrow \{3, 2, 1\}$ and $\{1, 2, 3\} \rightarrow \{2, 3, 1\}$, respectively. The permutations of rows, columns, and depth in Figure (b) are $\{1, 2, 3\} \rightarrow \{3, 2, 1\}$, $\{1, 2, 3\} \rightarrow \{2, 3, 1\}$, and $\{1, 2, 3\} \rightarrow \{3, 1, 2\}$, respectively.

figure 1(a), the 2D equivariant function demonstrates that when the indices rows and columns of the input matrix are permuted as $\{1, 2, 3\} \rightarrow \{3, 2, 1\}$ and $\{1, 2, 3\} \rightarrow \{2, 3, 1\}$, the output undergoes corresponding positional changes while maintaining the specific output features unaffected. Figure 1(b) depicts a 3D equivariant function, similar to the 2D case but additionally satisfying equivariance in the depth dimension. Building neural networks satisfying multi-dimensional equivariance is challenging as this property often involves tensors and has complex mathematical expressions. Linear layers with 1D equivariance were proposed in Zaheer et al. (2017) and extended to higher dimensions in Hartford et al. (2018). However, equivariant linear layers have shown inferior performance compared to nonlinear ones (Lee et al., 2019). Although a few studies have considered constructing 2D nonlinear equivariant layers (Maron et al., 2020; Duan et al., 2022), there is still a lack of an efficient framework for nonlinear networks that can be extended to higher-dimensional equivariance.

In this paper, we propose a novel serial multi-dimensional permutation equivariance (**SMPE**) framework for multi-dimensional equivariance, and the structure of its layers is shown in figure 2 (taking 3D equivariance as an example). Instead of directly designing networks based on multi-dimensional equivariance, we adopt a serial composition of 1D equivariant operations to achieve interactions among all-dimensional features while preserving multi-dimensional equivariance. The novelty lies in enhancing network depth through the serial composition of operations and strengthening network expressivity through the introduction of cross-dimensional global information and a feature reuse mechanism. Our key contributions comprise:

- We provide the first exact algebra-based definition of multi-dimensional permutation equivariance. Building upon this, we put forward the proposition for the composition of 1D permutation equivariant mappings, paving the way for the design of multi-dimensional permutation equivariant networks.

- Based on the proposed propositions, we build the multi-dimensional equivariant SMPE framework. Serial 1D equivariant operations in such a framework increase the network depth, while retaining a concise and easily understandable architecture. The introduction of cross-dimensional global information and feature reuse mechanisms endow the network with a strong expressivity. Besides, the SMPE framework can be easily extended to cases of invariance as well as the hybrid equivariance and invariance.

- We use an extensive set of experiments to evaluate the framework on contextual auction design, pseudo inverse computation, and typical wireless communication tasks. On such 2D and 3D equivariance-related tasks, our framework significantly outperforms the current state-of-the-art equivariant networks. For example, when computing the pseudo-inverse of

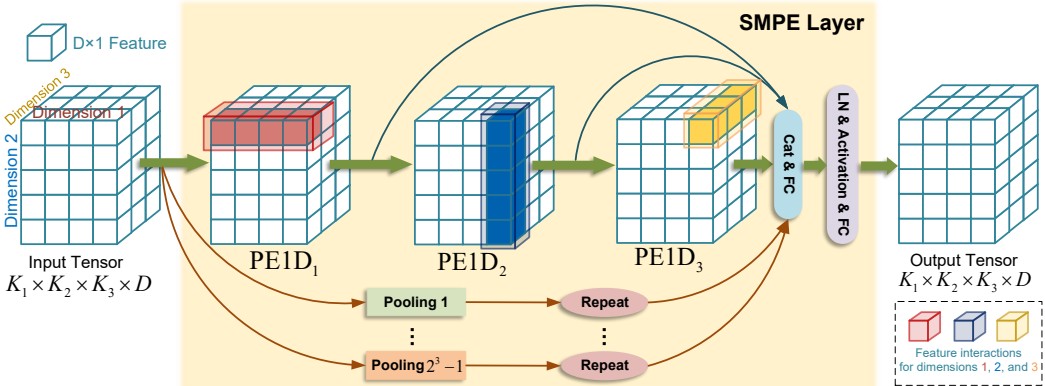

Figure 2: Single layer of SMPE framework consists of three parts (using 3D equivariance as an example in this figure): (i) Perform 1D equivariant operations on $N$ different dimensions serially. (ii) Use weighted connections to aggregate the output features that have been interacted across different dimensions. (iii) Introduce pooling operations to preserve cross-dimensional global information.

$7 \times 8$ matrices, the proposed framework reduces the mean absolute error (MAE) loss of other equivariant models by 76.6%. In two typical wireless communication tasks, SMPE could gain improvements as high as 10.6% and 7.1% over the state-of-the-art methods. Additionally, when the size of the set varies, the network can still operate and maintain excellent performance for slight size variation. For instance, a network trained on $8 \times 9$ matrices can achieve an MAE loss of $6.10 \times 10^{-2}$ when computing the pseudo inverse of $9 \times 10$ matrices.

## 2 RELATED WORK

### 2.1 ONE-DIMENSIONAL PERMUTATION EQUIVARIANT NETWORK

**Linear layers:** As the pioneering work, Zaheer et al. (2017) established the paradigms that permutation invariant networks must adhere to. Additionally, the paper introduced the forms of equivariant linear layers and further presented permutation invariant and equivariant networks. Based on this, Segol & Lipman (2019) and Vignac et al. (2020) proposed a simple universal equivariant network and a graph neural network with structural message-passing, respectively. Furthermore, Pan & Kondor (2022) extended the equivariant linear layers (Zaheer et al., 2017) into high-order interaction layers, and Maron et al. (2019) considered the invariant linear layer for high-order interaction. Similarly, a high-order equivariant graph variational encoder was proposed in Thiede et al. (2020). It is worth noting that despite its name similarity to high-dimensional equivariance, the high-order interaction (or high-order equivariance) is based on 1D equivariance and relies on the objects within a single set.

**Nonlinear layers:** Apart from linear equivariant layers, there are also some nonlinear layers primarily based on the transformer. With the transformer block without positional encodings, Lee et al. (2019) constructed nonlinear equivariant layers and pooling functions, based on which an invariant network is developed. Such a nonlinear network exhibits superior performance compared to networks composed of equivariant linear layers with activation functions. As theoretical support, Yun et al. (2019) proved that the transformer without positional encodings is a universal approximator for equivariant functions. Kim et al. (2021) combined the linear equivariant layers (Zaheer et al., 2017) with the transformer to obtain an equivariant transformer and extended it to the high-order interaction layers for hypergraph tasks.

**Applications:** In addition to these theoretical studies, some applications of equivariant networks have garnered attention. Using structures similar to linear equivariant layers, Qi et al. (2017) and Li & Lee (2019) addressed the tasks about point cloud and pose estimation, respectively. Leveraging

the 1D equivariance of the transformer variant, the network proposed by Sun et al. (2020) worked well on tasks such as robust line fitting and point cloud classification. Based on the variable node attention pooling mechanism, Meltzer et al. (2019) constructed a one-dimensional invariant network to effectively tackle graph classification problems. This issue was further thoroughly explored by Yang et al. (2022) through the hierarchical assembly of substructures and the help of soft sequence with context attention.

Nevertheless, unlike our proposed SMPE framework, the 1D equivariant (or invariant) neural networks mentioned above are not suitable for modeling multi-dimensional equivariant mappings.

## 2.2 Multi-Dimensional Permutation Equivariant Network

Hartford et al. (2018) demonstrated that linear layers satisfying multi-dimensional equivariance exhibit parameter sharing and proposed the structure of corresponding equivariant linear layers. Furthermore, Maron et al. (2018) put forward the linear layer for high-order interactions with multi-dimensional equivariance, which can degenerate into the structure presented in Hartford et al. (2018). Considering the design of 2D equivariant networks, Maron et al. (2020) investigates the relationship between the universality of the 1D equivariant block and the universality of the overall network. Unlike linear layers, the intricate nature of nonlinear layer construction poses a challenging task when extending them to multi-dimensional equivariant architectures. Therefore, there are only a few application-oriented works. For instance, a 2D equivariant network was designed by Umagami et al. (2023) to address prediction problems with table data. Duan et al. (2022) proposed a 2D equivariant network for contextual auction design. However, neither of these works considers the definition of higher-dimensional equivariance or the design of related nonlinear networks, which are tackled by our SMPE framework.

## 3 Method

In this section, we describe our SMPE framework. First, we provide the preliminaries. Then, we define the multi-dimensional permutation equivariance and build the SMPE framework. Finally, we extend our framework to encompass scenarios involving multi-dimensional permutation invariance.

## 3.1 Preliminaries

As a foundation for multi-dimensional permutation equivariance, we begin by introducing the definition of the symmetric group and permutation equivariance (Artin, 2011).

**Definition 3.1** The set of all bijections from the indices set $\mathbb{K} = \{1, 2, ..., K\}$ to $\mathbb{K}$ is called the *symmetric group*, denoted by $\mathbb{S}_K$ with cardinality $K!$.

Denote $\boldsymbol{X} \in \mathbb{R}^{K \times D_X}$ as the input matrix, where $K$ is the number of objects, and $D_X$ is the feature dimension. A permutation $\pi_K \in \mathbb{S}_K$ operates on $\mathbf{X}$ by permuting its item order: $(\pi_K \circ \boldsymbol{X})_{[k,:]} = \boldsymbol{X}_{[\pi_K^{-1}(k),:]}$, $\forall k \in \mathbb{K}$, where $\boldsymbol{X}_{[k,:]}$ represents the $k$-th row of $\boldsymbol{X}$.

**Definition 3.2** A mapping $f : \mathbb{R}^{K \times D_X} \to \mathbb{R}^{K \times D_Y}$ is said to be *permutation equivariant* if $f(\pi_K \circ \boldsymbol{X}) = \pi_K \circ f(\boldsymbol{X})$, $\forall \pi_K \in \mathbb{S}_K$.

## 3.2 SMPE Framework with Multi-Dimensional Equivariance

In this subsection, we consider the design of the networks that input the tensor $\mathbf{X} \in \mathbb{R}^{K_1 \times K_2 \times \cdots \times K_N \times D_X}$ and outputs $\mathbf{Y} \in \mathbb{R}^{K_1 \times K_2 \times \cdots \times K_N \times D_Y}$, where $K_n$ is the number of objects in the $n$-th dimension, and $D_X$ is the feature dimension. Such a network is expected to facilitate interactions among features across all dimensions while maintaining $N$-dimensional equivariance.

We first define the multi-dimensional symmetric group and permutation equivariance.

**Definition 3.3** We define the *$N$-dimensional symmetric group* as $\mathbb{S}^{\mathbb{N}} = \mathbb{S}_{K_1} \times \mathbb{S}_{K_2} \times \cdots \times \mathbb{S}_{K_N}$, whose cardinality is $\Pi_{n=1}^{N}(K_n!)$, where $\mathbb{N} = \{1, 2, ..., N\}$.

Note that the superscript of $\mathbb{S}$ denotes the dimensions to which the symmetric group is applied. For instance, $\mathbb{S}^{\{1,3,5\}} = \mathbb{S}_{K_1} \times \mathbb{S}_{K_3} \times \mathbb{S}_{K_5}$ operates on the first, third, and fifth dimension of $\mathbf{X}$. A

permutation $\pi^{\mathbb{N}} = (\pi_{K_1}, \pi_{K_2}, ..., \pi_{K_N}) \in \mathbb{S}^{\mathbb{N}}$ operates on $\mathbf{X}$ by respectively permuting the order of objects along $1, ..., N$-th dimension with

$$(\pi^{\mathbb{N}} \circ \mathbf{X})_{[k_1, k_2, ..., k_N, :]} = \mathbf{X}_{[\pi_{K_1}^{-1}(k_1), \pi_{K_2}^{-1}(k_2), ..., \pi_{K_N}^{-1}(k_N), :]}, \ \forall k_1 \in \mathbb{K}_1, ..., k_N \in \mathbb{K}_N, \quad (1)$$

where $\mathbb{K}_n = \{1, ..., K_n\}, \forall n \in \mathbb{N}$.

**Definition 3.4** A mapping $f : \mathbb{R}^{K_1 \times K_2 \times \cdots \times K_N \times D_X} \rightarrow \mathbb{R}^{K_1 \times K_2 \times \cdots \times K_N \times D_Y}$ is said to be $N$-*dimensional permutation equivariant* if $f(\pi^{\mathbb{N}} \circ \mathbf{X}) = \pi^{\mathbb{N}} \circ f(\mathbf{X}), \ \forall \pi^{\mathbb{N}} \in \mathbb{S}^{\mathbb{N}}$.

A 1D equivariant network is designed to maintain the same output features for their corresponding inputs when the order of objects is permuted. When considering multi-dimensional equivariance, the network needs to satisfy this property across multiple dimensions, making it more challenging to design and resulting in a complicated pattern of parameter sharing Hartford et al. (2018). To address this issue, we try to merge 1D equivariant networks to construct the required high-dimensional equivariant network. Following this idea, we first put forward the following proposition

**Proposition 3.1** Let $h_n : \mathbb{R}^{K_n \times D} \rightarrow \mathbb{R}^{K_n \times D}$ denote the 1D equivariant mapping for $1 \leq n \leq N$. Additionally, define the function $\bar{h}_n : \mathbb{R}^{K_1 \times \cdots \times K_N \times D} \rightarrow \mathbb{R}^{K_1 \times \cdots \times K_N \times D}$ as the operation that applies the function $h_n$ to the $n$-th dimension of $\mathbf{X}$, while keeping an identity mapping for the remaining $N - 1$ dimensions[1], then

(i) The function $g = \bar{h}_{n_1} \circ \bar{h}_{n_2} \circ \cdots \circ \bar{h}_{n_M}, 1 \leq n_m \leq N, \forall m = 1, ..., M$ exhibits $N$-dimensional permutation equivariance.

(ii) The function $g = \bar{h}_{n_1} + \bar{h}_{n_2} + \cdots + \bar{h}_{n_M}, 1 \leq n_m \leq N, \forall m = 1, ..., M$ exhibits $N$-dimensional permutation equivariance.

The above proposition shows that the network composed of 1D equivariant networks through addition or composition possesses multi-dimensional equivariance.

Subsequently, we consider endowing the network with the ability to facilitate interactions among all dimensions. Denote $f_n : \mathbb{R}^{K_n \times D} \rightarrow \mathbb{R}^{K_n \times D}, 1 \leq n \leq N$ as the 1D equivariant layer operating on the $n$-th dimension of $\mathbf{X}$. Different from $h_n$, $f_n$ is designed to perform interactions among $K_n$ object features, such as the layers developed by Zaheer et al. (2017) and Lee et al. (2019). Similar to $\bar{h}_n$, we define $\bar{f}_n$ based on $f_n$. It can be easily verified that if the network is composed of $\{\bar{f}_n\}_{n=1}^N$, then it enables interactions among features across all $N$ dimensions. According to Proposition 3.1, the two simplest combinations are in a serial and parallel manner, expressed as

$$g_{\text{serial}} = \bar{f}_N \circ \bar{f}_{N-1} \circ \cdots \circ \bar{f}_1, \ g_{\text{para}} = \bar{f}_N + \bar{f}_{N-1} + \cdots + \bar{f}_1. \quad (2)$$

Although $g_{\text{serial}}$ and $g_{\text{para}}$ have the same computational complexity, $g_{\text{serial}}$ has a deeper network depth compared to $g_{\text{para}}$ due to its sequential processing, which means $g_{\text{serial}}$ may have a stronger expressivity (Lu et al., 2017). Thus, we choose $g_{\text{serial}}$ as the prototype of the framework.

Since each application of $\bar{f}_1, ..., \bar{f}_N$ to the input represents the interactions between the features in a new dimension, the output features of $\bar{f}_1, ..., \bar{f}_N$ will exhibit different distributions, respectively. Therefore, we utilize the weighted connections to aggregate the outputs of each $\bar{f}_n$ for feature reuse (Huang et al., 2017). Furthermore, we highlight the significance of global information and the limitations of $g_{\text{serial}}$ in this context. When dealing with elements in a set, their global information is crucial (e.g., mean or max), as validated by Zaheer et al. (2017). Extending 1D equivariance to multi-dimensional equivariance requires the global information not only within individual dimensions but also across multiple dimensions Hartford et al. (2018). However, $g_{\text{serial}}$ may lose the global information for two reasons: (i) If $f_n$ is nonlinear (e.g., the layer with attention mechanism), preserving global information within

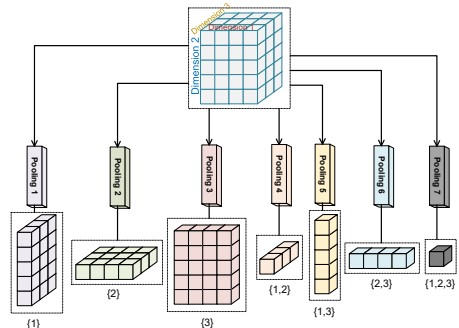

Figure 3: Pooling operations for $N = 3$.

---

[1]In practical implementation, other $N - 1$ dimensions can be considered as batch dimensions.

a single dimension becomes challenging. (ii) Applying permutation equivariant operations separately to individual dimensions hinders the preservation of joint global information across multiple dimensions. To this end, we add the weighted global information to the output (He et al., 2016). In summary, the SMPE framework can be represented as

$$g_{\text{SMPE}}(\mathbf{X}) = \sigma \left( \sum_{n=1}^{N} w_n^{\text{PE}} [\bar{f}_n \circ \bar{f}_{n-1} \circ \cdots \circ \bar{f}_1(\mathbf{X})] + \sum_{\mathbb{P} \subseteq \mathbb{N} \setminus \{\emptyset\}} w_{\mathbb{P}}^{\text{GI}} \bar{\mathbf{X}}_{\mathbb{P}} \right), \tag{3}$$

where $w_n^{\text{PE}}$ is the weight belonging to the output of $\bar{f}_n$, $w_{\mathbb{P}}^{\text{GI}}$ is the weight belonging to the global information $\bar{\mathbf{X}}_{\mathbb{P}}$, $\sigma$ is the element-wise nonlinearity, and $\mathbb{N} = \{1, 2, ..., N\}$. $\bar{\mathbf{X}}_{\mathbb{P}} \in \mathbb{R}^{K_1 \times \cdots \times K_N \times D}$, $\mathbb{P} \subseteq \mathbb{N} \setminus \{\emptyset\}$ is obtained by pooling operations applied at dimension pattern $\mathbb{P}$ (e.g., the mean and max functions). In the case of $N = 3$, for example, the power set of $\mathbb{N} \setminus \{\emptyset\}$ is $\{\{1\}, \{2\}, \{3\}, \{1, 2\}, \{1, 3\}, \{2, 3\}, \{1, 2, 3\}\}$, which are shown in Figure 3. After the pooling operation, the input is then repeated to restore its original shape. Note that the number of subsets of set $\mathbb{N} \setminus \{\emptyset\}$ is $2^N - 1$. The following proposition validates the versatility of the SMPE framework.

**Proposition 3.2** The SMPE framework in equation 3 will degenerate to the exchangeable tensor layer (the combination of the multi-dimensional equivariant linear layer and element-wise nonlinearity in Hartford et al. (2018)) when $\bar{f}_n = f_{\text{id}}$, $n = 1, ..., N$, where $f_{\text{id}}$ is the identity mapping $f_{\text{id}}(\mathbf{X}) = \mathbf{X}$.

Then, we describe the implementation details of the SMPE layer. Without loss of generality, we consider the same input and output dimensions $D$ [2]. The global information $\bar{\mathbf{X}}$ and the outputs $\mathbf{O}$ from each layer are concatenated and then linearly combined using linear layers $\text{FC}_2 : \mathbb{R}^{(N+2^N-1)D} \to \mathbb{R}^D$ applied at the last dimension. Besides, we employ layer normalization LN before the activation function $\sigma$ (Ba et al., 2016). The architecture of SMPE layer $\text{SMPEL} : \mathbb{R}^{K_1 \times \cdots \times K_N \times D} \to \mathbb{R}^{K_1 \times \cdots \times K_N \times D}$ can be represented as follows:

$$\text{SMPEL}(\mathbf{X}) = \text{FC}_1(\sigma(\text{LN}(\mathbf{M}))), \tag{4}$$

$$\mathbf{M} = \text{FC}_2([\mathbf{O}_1, \mathbf{O}_2, ..., \mathbf{O}_N, \bar{\mathbf{X}}_{\mathbb{P}_1}, ..., \bar{\mathbf{X}}_{\mathbb{P}_{2^N-1}}]), \tag{5}$$

$$\mathbf{O}_0 = \mathbf{X}, \ \mathbf{O}_n = \text{PE1D}_n(\mathbf{O}_{n-1}), \ n = 1, 2, ..., N, \tag{6}$$

where $\text{FC}_1 : \mathbb{R}^D \to \mathbb{R}^D$ represents the linear layer applied at the last dimension, and $\text{PE1D}_n : \mathbb{R}^{K_n \times D} \to \mathbb{R}^{K_n \times D}$ denotes the 1D equivariant layer operating at the $n$-th dimension. The architecture of the SMPE layer is illustrated in Figure 2, where $\text{PE1D}_1$, $\text{PE1D}_2$, and $\text{PE1D}_3$ are performed at the first, second, and third dimensions. By stacking SMPEL, we can build the multi-dimensional equivariant network SMPEN.

## 3.3 EXTENTION TO PERMUTATION INVARIANCE

Permutation invariant neural networks are important tools for capturing global features from sets or multiple sets Chien et al. (2021); Wagstaff et al. (2019), such as table understanding (Dash et al., 2022) and parameter estimation (Zaheer et al., 2017). While the SMPE framework is proposed for multi-dimensional equivariance, it can also handle multi-dimensional invariance. We first provide the definition of permutation invariance and extend it to the cases with higher dimensions.

**Definition 3.5** A mapping $f : \mathbb{R}^{K \times D_X} \to \mathbb{R}^{D_Y}$ is said to be *permutation invariant* if $f(\pi_K \circ \boldsymbol{X}) = f(\boldsymbol{X})$, $\forall \pi \in \mathbb{S}_K$.

**Definition 3.6** A mapping $f : \mathbb{R}^{K_1 \times \cdots \times K_N \times D_X} \to \mathbb{R}^{D_Y}$ is said to be *N-dimensional permutation invariant* if $f(\pi^{\mathbb{N}} \circ \mathbf{X}) = f(\mathbf{X})$, $\forall \pi^{\mathbb{N}} \in \mathbb{S}^{\mathbb{N}}$.

According to Zaheer et al. (2017), with the help of pooling operations, 1D permutation equivariant functions can be transformed into permutation invariant functions. These pooling operations contain functions like mean, max, and sum operations. It can also be constructed by a neural network, such as the pooling layer in Lee et al. (2019). We extend this idea to higher dimensions. Following this idea, the serial multi-dimensional permutation invariance network (SMPIN) can be expressed as

$$\text{SMPIN}(\mathbf{X}) = \text{Pool}_{\mathbb{N}}(\text{SMPEN}(\mathbf{X})), \tag{7}$$

---

[2]The network can use linear layers to map the feature dimension $D_X$ to feature dimension $D$ in the hidden layer, and subsequently map $D$ to $D_Y$.

where $\text{Pool}_{\mathbb{N}}$ represents the pooling operation on dimensions in $\mathbb{N}$. Furthermore, we consider the case where the mapping needs to satisfy equivariance in some dimensions and invariance in others when dealing with multi-dimensional tensor inputs.

**Definition 3.7** A mapping $f : \mathbb{R}^{K_1 \times \cdots \times K_N \times D_X} \to \mathbb{R}^{K_1 \times \cdots \times K_M \times D_Y}$, $1 \leq M \leq N - 1$ is said to be $M$-$(N-M)$-*dimensional hybrid permutation equivariant and invariant if*

$$f\left(\pi^{\mathbb{M}^-} \circ \mathbf{X}\right) = \pi^{\mathbb{M}^-} \circ f(\mathbf{X}), \ \forall \pi^{\mathbb{M}^-} \in \mathbb{S}^{\mathbb{M}^-}, \ f\left(\pi^{\mathbb{M}^+} \circ \mathbf{X}\right) = f(\mathbf{X}), \ \forall \pi^{\mathbb{M}^+} \in \mathbb{S}^{\mathbb{M}^+}, \quad (8)$$

where $\mathbb{M}^- = \{1, 2, ..., M\}$ and $\mathbb{M}^+ = \{M + 1, ..., N\}$.

We provide a simple example to illustrate the mappings that satisfy the properties mentioned above. $M$ different systems input the same random signal and output $M$ types of signals with different distributions. Estimating the parameters of these $M$ distributions from $K$ discrete samples each is a mapping $f : \mathbb{R}^{S \times K \times D_X} \to \mathbb{R}^{S \times D_Y}$, which satisfies 1-1-dimensional (1-1D) hybrid permutation equivariance and invariance. Note that this problem is not reasonable to be divided into $S$ separate parameter estimation tasks $f : \mathbb{R}^{K \times D_X} \to \mathbb{R}^{D_Y}$, as the $S$ different output signals share similar characteristics due to the common input. Similar to equation 7, the serial multi-dimensional hybrid permutation equivariant and invariant network (SMPEIN) can be expressed as

$$\text{SMPEIN}(\mathbf{X}) = \text{Pool}_{\mathbb{M}^+} \left(\text{SMPEN}(\mathbf{X})\right). \quad (9)$$

## 4 Experiment

Three types of experiments are considered in this section: (1) pseudo inverse computation, (2) transmission design for multi-input multi-output (MIMO) system, (3) transmission design for cell-free MIMO system, where the experiments (1) and (2) contain the objects across two dimensions, and experiment (3) contains the objects across three dimensions. To show the performance of the 1D equivariant network in handling multi-dimensional equivariant problems, we adopted the approach in Lee et al. (2019) as one of the baselines. The network composed of multi-dimensional equivariant linear layers in Hartford et al. (2018) is also compared. Umagami et al. (2023) and Duan et al. (2022) proposed 2D equivariant network structures, which are both based on the transformer and share similar structures. Since Umagami et al. (2023) considers the time prediction in another dimension that deviates from the focus of this paper, we selected the approach presented by Duan et al. (2022) as another baseline. We refer to the 1D network in Lee et al. (2019), the 2D network in Duan et al. (2022), the 2D and 3D networks in Hartford et al. (2018) as `SetTrans1D`, `CITransNet2D`, `ETN2D`, and `ETN3D`, respectively. The 2D and 3D networks constructed by SMPE using the deepset layer (Zaheer et al., 2017) and transformer encoding layer (Vaswani et al., 2017) as $f_n$ are denoted as `SMPEN2D−DS`, `SMPEN2D−TF`, `SMPEN3D−DS`, and `SMPEN3D−TF`. Due to space limitations, details of tasks and related experiments are not provided in this section. Please refer to Appendices D and E for more information.

### 4.1 Pseudo Inverse Computation

Pseudo inverse is a generalization of matrix inverse (Courrieu, 2008), and it is an essential matrix operation in many fields of research. Networks in this subsection are trained to compute the transposed pseudo inverse $(\mathbf{X}^\dagger)^T = (\mathbf{X}^H(\mathbf{X}\mathbf{X}^H)^{-1})^T \in \mathbb{C}^{K \times N}$ of the input matrix $\mathbf{X} \in \mathbb{C}^{K \times N}, K < N$. It is easy to prove that pseudo inverse computation is a 2D permutation equivariant function. We use the mean absolute error (MAE) loss and mean squared error (MSE) loss between the result and $(\mathbf{X}^\dagger)^T$ as two evaluation metrics. Each evaluation metric is separately computed and summed for the real and imaginary parts.

The comparisons of MAE and MSE for three different configurations of $K$ and $N$ are shown in Table 1. It can be observed that due to the differences between 1D and 2D equivariance, the performance of `SetTrans1D` is lower than that of other networks. Compared with other 2D equivariant networks, `SMPEN2D−DS` and `SMPEN2D−TF` have the most superior performance. For example, when computing pseudo inverses of $8 \times 9$ matrices, `SMPEN2D−DS` and `SMPEN2D−TF` have an MAE loss that is 13.9% and 70.9% lower than `ETN2D`, respectively. Table 1 also compares the performance of the networks when trained in multiple scenarios but tested in one scenario. The results indicate that equivariant networks exhibit the generalization to work with varying numbers of objects.

Table 1: MAE and MSE ($10^{-2}$, the lower the better) performance comparisons about computing transposed pseudo inverse.

| | Test Setting | $\mathcal{A}$: $K = 7, N = 8$ | | | $\mathcal{B}$: $K = 8, N = 9$ | | | $\mathcal{C}$: $K = 9, N = 10$ | | |
|---|---|---|---|---|---|---|---|---|---|---|
| | Train Setting | $\mathcal{A}$ | $\mathcal{B}$ | $\mathcal{C}$ | $\mathcal{A}$ | $\mathcal{B}$ | $\mathcal{C}$ | $\mathcal{A}$ | $\mathcal{B}$ | $\mathcal{C}$ |
| **MAE($\downarrow$)** | SetTrans1D | 16.12 | 16.14 | 16.19 | 15.46 | 15.43 | 15.45 | 14.81 | 14.74 | 14.73 |
| | ETN2D | 9.76 | 10.71 | 11.98 | 11.31 | 10.98 | 11.56 | 12.55 | 11.65 | 11.62 |
| | CITransNet2D | 9.51 | 10.33 | 11.94 | 11.64 | 10.98 | 12.05 | 12.57 | 11.67 | 12.09 |
| | SMPEN2D$-$DS | 7.60 | 9.55 | 11.22 | 10.34 | 9.45 | 10.37 | 12.16 | 10.69 | 10.56 |
| | SMPEN2D$-$TF | **2.23** | 5.70 | 12.81 | 6.13 | **3.20** | 6.41 | 9.41 | 6.10 | **4.59** |
| | Test Setting | $\mathcal{A}$: $K = 7, N = 8$ | | | $\mathcal{B}$: $K = 8, N = 9$ | | | $\mathcal{C}$: $K = 9, N = 10$ | | |
| | Train Setting | $\mathcal{A}$ | $\mathcal{B}$ | $\mathcal{C}$ | $\mathcal{A}$ | $\mathcal{B}$ | $\mathcal{C}$ | $\mathcal{A}$ | $\mathcal{B}$ | $\mathcal{C}$ |
| **MSE($\downarrow$)** | SetTrans1D | 5.35 | 5.36 | 5.38 | 5.05 | 5.04 | 5.05 | 4.43 | 4.40 | 4.40 |
| | ETN2D | 3.27 | 3.96 | 5.21 | 3.67 | 3.85 | 4.21 | 3.68 | 3.59 | 3.55 |
| | CITransNet2D | 3.42 | 3.48 | 3.91 | 3.76 | 3.65 | 3.95 | 3.61 | 3.44 | 3.60 |
| | SMPEN2D$-$DS | 2.22 | 3.39 | 5.20 | 3.04 | 3.14 | 3.63 | 3.30 | 3.06 | 3.05 |
| | SMPEN2D$-$TF | **0.43** | 1.28 | 4.18 | 1.56 | **0.83** | 2.10 | 2.26 | 1.31 | **0.98** |

Equivariant networks can still maintain good performance when there is little discrepancy between the training and testing scenarios. Besides, although the performance of SetTrans1D is relatively worse, it exhibits more robust generalization as its performance remains nearly the same under three different training cases.

## 4.2 TRANSMISSION DESIGN FOR MIMO SYSTEM

As a crucial foundation for modern human society, the tasks in wireless communication systems often involve higher-dimensional permutation equivariant mappings Wang et al. (2023). The reason is that communication systems often contain multiple sets like multiple users, each equipped with multiple antennas, and several base stations, each with multiple antennas. For instance, multi-user (MU)-multiple-input-single-output (MISO) (Zhao et al., 2022), MU-MIMO (Christensen et al., 2008), and cell-free MU-MIMO systems (Feng et al., 2021) contain two, three, and four sets of objects, respectively. To demonstrate the advantage of our proposed framework in modeling high-dimensional equivariant mappings, we consider two typical communication tasks in this subsection and the following one, each involving two and three sets of objects, respectively.

Consider a MIMO downlink system where an $N$-antenna base station (BS) transmits the signal to $K$ single-antenna user equipment (UE). The networks are trained to design an optimal transmission scheme $\widetilde{\mathbf{W}} \in \mathbb{R}^{K \times N \times 2}$ based on the available information $\widetilde{\mathbf{H}} \in \mathbb{R}^{K \times N \times 3}$, i.e., $f(\widetilde{\mathbf{H}}) = \widetilde{\mathbf{W}}^{\star}$, which is a 2D equivariant mapping. The evaluation metric $\bar{R}_{\mathrm{M}}$ is the average rate of MIMO transmission with different signal-to-noise ratios (SNRs).

Table 2: Transmission performance $\bar{R}_{\mathrm{M}}$ (the higher the better) comparisons for MIMO system.

| | Test Setting | $\mathcal{A}$: $N = 7, K = 7$ | | | $\mathcal{B}$: $N = 8, K = 7$ | | | $\mathcal{C}$: $N = 8, K = 8$ | | |
|---|---|---|---|---|---|---|---|---|---|---|
| | Train Setting | $\mathcal{A}$ | $\mathcal{B}$ | $\mathcal{C}$ | $\mathcal{A}$ | $\mathcal{B}$ | $\mathcal{C}$ | $\mathcal{A}$ | $\mathcal{B}$ | $\mathcal{C}$ |
| $\bar{R}_{\mathrm{M}}(\uparrow)$ | SetTrans1D | 7.30 | 7.30 | 7.30 | 8.00 | 8.00 | 8.00 | 8.16 | 8.16 | 8.16 |
| | ETN2D | 16.76 | 17.83 | 14.98 | 17.40 | 18.76 | 15.68 | 14.08 | 15.35 | 19.16 |
| | CITransNet2D | 20.55 | 20.56 | 20.34 | 21.73 | 21.90 | 21.64 | 21.96 | 22.10 | 22.07 |
| | SMPEN2D$-$DS | 17.89 | 19.53 | 15.92 | 18.61 | 20.68 | 16.65 | 14.79 | 16.14 | 20.71 |
| | SMPEN2D$-$TF | **22.21** | 22.70 | 20.69 | 23.78 | **24.71** | 22.30 | 21.80 | 23.17 | **24.44** |

Table 2 presents the comparison of $\bar{R}_{\mathrm{M}}$ performance. As it can be observed, the performance of the proposed SMPEN2D$-$DS outperforms that of ETN2D, and the performance of SMPEN2D$-$TF surpasses that of CITransNet2D. This difference is most prominent when $N = 8, K = 7$, where

SMPEN2D$-$DS achieves a 10.2% higher performance than ETN2D, and SMPEN2D$-$TF achieves a 12.8% higher performance than CITransNet2D.

## 4.3 TRANSMISSION DESIGN FOR CELL-FREE MIMO SYSTEM

We further consider the transmission design for cell-free MIMO systems (Feng et al., 2022). There is a central processing unit (CPU) controlling $M$ distributed access points (APs) to support $K$ single-antenna UEs, where each AP is equipped with $N$ antennas. The networks are trained to design an optimal transmission scheme $\widetilde{\mathbf{W}} \in \mathbb{R}^{M \times K \times N \times 2}$ based on the available information $\widetilde{\mathbf{H}} \in \mathbb{R}^{M \times K \times N \times 3}$, i.e., $f(\widetilde{\mathbf{H}}) = \widetilde{\mathbf{W}}^{\star}$, which is a 3D equivariant mapping. The evaluation metric $\bar{R}_{\text{CFM}}$ is the average rate of cell-free MIMO transmission with different SNRs.

Table 3: Transmission performance $\bar{R}_{\text{CFM}}$ (the higher the better) comparisons for cell-free MIMO system.

| | **Test Setting** | $\mathcal{A}$: $M\!=\!3, K\!=\!6, N\!=\!2$ | | | $\mathcal{B}$: $M\!=\!2, K\!=\!6, N\!=\!3$ | | | $\mathcal{C}$: $M\!=\!4, K\!=\!8, N\!=\!2$ | | |
|---|---|---|---|---|---|---|---|---|---|---|
| | **Train Setting** | $\mathcal{A}$ | $\mathcal{B}$ | $\mathcal{C}$ | $\mathcal{A}$ | $\mathcal{B}$ | $\mathcal{C}$ | $\mathcal{A}$ | $\mathcal{B}$ | $\mathcal{C}$ |
| | CITransNet2D | 19.40 | 18.70 | 19.00 | 18.20 | 17.90 | 18.20 | 22.60 | 20.20 | 24.30 |
| | ETN2D | 15.90 | 16.30 | 10.40 | 14.80 | 15.50 | 9.87 | 11.40 | 11.50 | 20.90 |
| $\bar{R}_{\text{CFM}}(\uparrow)$ | ETN3D | 17.29 | 14.20 | 10.40 | 13.58 | 15.80 | 9.11 | 13.72 | 11.50 | 19.90 |
| | SMPEN2D$-$DS | 15.90 | 16.60 | 9.01 | 14.90 | 16.00 | 8.42 | 11.00 | 12.60 | 19.20 |
| | SMPEN3D$-$DS | 20.10 | 12.20 | 10.50 | 12.20 | 18.80 | 8.27 | 16.20 | 12.30 | 22.50 |
| | SMPEN2D$-$TF | 20.80 | 19.40 | 19.60 | 19.50 | 18.60 | 18.50 | 17.20 | 21.30 | 26.80 |
| | SMPEN3D$-$TF | **23.80** | 9.66 | 18.90 | 10.60 | **23.30** | 9.50 | 20.10 | 9.96 | **30.40** |

Table 3 provides $\bar{R}_{\text{CFM}}$ performance comparison of testing under one setting and training across multiple settings. Interestingly, although the 3D networks (light blue background) perform well when testing on their training cases, their performance drops more significantly when tested in other cases compared to the 2D networks. The reason is that the 3D networks exhibit more evident changes in the number of objects relative to the 2D network. For example, from case $\mathcal{A}$ to $\mathcal{B}$, the tensor dimension changes for 2D networks is $\mathbf{X} \in \mathbb{R}^{6 \times 6 \times 3} \rightarrow \mathbf{X} \in \mathbb{R}^{6 \times 6 \times 3}$. The change for the 3D network is $\mathbf{X} \in \mathbb{R}^{3 \times 6 \times 2} \rightarrow \mathbf{X} \in \mathbb{R}^{2 \times 6 \times 3}$, where $M$ changes from 3 to 2, and $N$ changes from 2 to 3. It can be concluded from Table 1 that while equivariant networks can still work as the number of objects changes, the performance degradation becomes more evident as object quantity changes more significantly. Therefore, 3D networks do not generalize as well to changes in tensor dimensions compared to the 2D network. This conclusion is also supported by the generalization performance of SetTrans1D in Table 1 and Table 2.

## 5 CONCLUSION AND OUTLOOK

In this paper, we proposed a multi-dimensional permutation equivariant framework called SMPE. Concretely, we first provide the algebraic definition of multi-dimensional permutation equivariance. Building upon this, we proposed the proposition that combining 1D equivariant functions applied to individual dimensions can still preserve higher-dimensional equivariance. By composing 1D equivariant operations that act on different dimensions, we proposed the SMPE framework, which possesses multi-dimensional equivariance and can facilitate interactions among objects in all dimensions. This framework utilizes the feature reuse mechanism and introduces cross-dimensional global information to enhance the expressivity. Besides, with the help of pooling operations, SMPE framework can be extended to the case of multi-dimensional invariance. Experiments demonstrated that the networks constructed by the SMPE framework exhibited superior performance and could operate on different sizes of sets. In future work, we will consider investigating training strategies to enhance the performance of equivariant networks on cases with different set sizes.

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

## A  PROOF OF PROPOSITION 3.1

In this section, we provide the proof of Proposition 3.1. The permutation $\pi^{\mathbb{N}}$ from $\mathbb{S}^{\mathbb{N}}$ can be decomposed into $\pi^{\mathbb{N}} = \pi_{K_N} \circ \pi_{K_{N-1}} \circ \cdots \circ \pi_{K_1}$, where $\pi_{K_n} \in \mathbb{S}_{K_n}$ is applied at the $n$-th dimension of the input tensor, and the order of permutations at different dimensions can be changed. Therefore, we have

$$
\begin{aligned}
\bar{h}_n(\pi^{\mathbb{N}} \circ \mathbf{X}) &= \bar{h}_n(\pi_{K_N} \circ \pi_{K_{N-1}} \circ \cdots \circ \pi_{K_1} \circ \mathbf{X}) \\
&= \bar{h}_n\left(\pi_{K_n} \circ (\pi_{K_N} \circ \cdots \circ \pi_{K_{n-1}} \circ \pi_{K_{n+1}} \circ \cdots \circ \pi_{K_1} \circ \mathbf{X})\right) \\
&= \pi_{K_n} \circ \bar{h}_n(\pi_{K_N} \circ \cdots \circ \pi_{K_{n-1}} \circ \pi_{K_{n+1}} \circ \cdots \circ \pi_{K_1} \circ \mathbf{X}), \ \forall \pi^{\mathbb{N}} \in \mathbb{S}^{\mathbb{N}}.
\end{aligned} \tag{10}
$$

Since $\bar{h}_n$ perform the identity mapping $h_{\mathrm{id}}(\mathbf{X}) = \mathbf{X}$ in other dimensions of $\mathbf{X}$, it exhibits permutation equivariance in other $N-1$ dimensions. Then, the above equation can be written as

$$
\begin{aligned}
\bar{h}_n(\pi^{\mathbb{N}} \circ \mathbf{X}) &= \pi_{K_n} \circ \bar{h}_n(\pi_{K_N} \circ \cdots \circ \pi_{K_{n-1}} \circ \pi_{K_{n+1}} \circ \cdots \circ \pi_{K_1} \circ \mathbf{X}) \\
&= \pi_{K_N} \circ \pi_{K_{N-1}} \circ \cdots \circ \pi_{K_1} \circ \bar{h}_n(\mathbf{X}) \\
&= \pi^{\mathbb{N}} \circ \bar{h}_n(\mathbf{X}), \ \forall \pi^{\mathbb{N}} \in \mathbb{S}^{\mathbb{N}}.
\end{aligned} \tag{11}
$$

Based on this equation, we can obtain the following conclusion

$$
\begin{aligned}
g(\pi^{\mathbb{N}} \circ \mathbf{X}) &= \bar{h}_{n_1} \circ \bar{h}_{n_2} \circ \cdots \circ \bar{h}_{n_M}(\pi^{\mathbb{N}} \circ \mathbf{X}) \\
&= \bar{h}_{n_1} \circ \bar{h}_{n_2} \circ \cdots \circ (\pi^{\mathbb{N}} \circ \bar{h}_{n_M}(\mathbf{X})) \\
&= \pi^{\mathbb{N}} \circ g(\mathbf{X}), \ \forall \pi^{\mathbb{N}} \in \mathbb{S}^{\mathbb{N}}.
\end{aligned} \tag{12}
$$

This completes the proof for Proposition 3.1 (i). For Proposition 3.1 (ii), we also have the following equation:

$$
\begin{aligned}
g(\pi^{\mathbb{N}} \circ \mathbf{X}) &= \bar{h}_{n_1}(\pi^{\mathbb{N}} \circ \mathbf{X}) + \bar{h}_{n_2}(\pi^{\mathbb{N}} \circ \mathbf{X}) + \cdots + \bar{h}_{n_M}(\pi^{\mathbb{N}} \circ \mathbf{X}) \\
&= \pi^{\mathbb{N}} \circ \bar{h}_{n_1}(\mathbf{X}) + \pi^{\mathbb{N}} \circ \bar{h}_{n_2}(\mathbf{X}) + \cdots + \pi^{\mathbb{N}} \circ \bar{h}_{n_M}(\mathbf{X}) \\
&= \pi^{\mathbb{N}} \circ g(\mathbf{X}), \ \forall \pi^{\mathbb{N}} \in \mathbb{S}^{\mathbb{N}},
\end{aligned}
\tag{13}
$$

which completes the proof for Proposition 3.1.

## B  PROOF OF PROPOSITION 3.2

In this section, we provide the proof of Proposition 3.2. Without loss of generality, we consider the case where $D = 1$ aligns with the expression in Hartford et al. (2018). The exchangeable tensor layer is expressed as $g_{\text{ETL}}(\mathbf{X}) = \text{vec}^{-1}\sigma(\boldsymbol{W}\text{vec}(\mathbf{X}))$, where $\boldsymbol{W} \in \mathbb{R}^{(\Pi_{n=1}^{N} K_n) \times (\Pi_{n=1}^{N} K_n)}$ is the trainable weight matrix with specific parameter sharing patterns. A distinct parameter $w_{\mathbb{S}}$ is defined for each $\mathbb{S} \subseteq \mathbb{N} = \{1, 2, ..., N\}$, and the elements of $\boldsymbol{W}$ are tied as follows

$$
\boldsymbol{W}_{[\boldsymbol{p},\boldsymbol{q}]} := w_{\mathbb{S}} \quad \text{s.t. } p_i = q_i, \ i \in \mathbb{S}, \ p_{i'} \neq q_{i'}, \ i' \in \mathbb{N}\backslash\mathbb{S},
\tag{14}
$$

where $\boldsymbol{p} \in \mathbb{R}^{N \times 1}$ and $\boldsymbol{q} \in \mathbb{R}^{N \times 1}$ are the row and column index represented by the $N$-dimensional coordinates for shape $K_1 \times K_2 \times \cdots \times K_N$. Besides, we have $1 \leq p_i \leq K_i, 1 \leq q_i \leq K_i, i = 1, ..., N$. Equation 14 implies that, for a specific set of dimensions $\mathbb{S}$, the elements of $\boldsymbol{W}$, which satisfy that the $N$-dimensional row coordinate $\boldsymbol{m}$ are the same as the column coordinate $\boldsymbol{n}$ on dimensions in $\mathbb{S}$, share the same weight $w_{\mathbb{S}}$. Due to $|\mathbb{N}| = N$, there are $2^N$ different elements $w_{\mathbb{S}}$ in $\boldsymbol{W}$. Then, we consider simplifying the above expression and derive a simple representation of $g_{\text{ETL}}(\mathbf{X})$. Define $\mathbf{M} = \text{vec}^{-1}(\boldsymbol{W}\text{vec}(\mathbf{X}))$, and we have $g_{\text{ETL}}(\mathbf{X}) = \sigma(\mathbf{M})$. The $p_1, ..., p_N$-th element of $\mathbf{M}$ is given by

$$
\begin{aligned}
& m_{p_1,...,p_N} \\
&= \boldsymbol{W}_{[\boldsymbol{p},:]} \times \text{vec}(\mathbf{X}) \\
&= \sum_{q_N=1}^{K_N} \sum_{q_{N-1}=1}^{K_{N-1}} \cdots \sum_{q_1=1}^{K_1} \boldsymbol{W}_{[\boldsymbol{p},(q_1,q_2,...,q_N)^T]} \cdot x_{q_1,...,q_N} \\
&= \sum_{\mathbb{S} \subseteq \mathbb{N}} w_{\mathbb{S}} \sum_{\substack{q_i = p_i, i \in \mathbb{S} \\ q_{i'} \neq p_{i'}, i' \in \mathbb{N}\backslash\mathbb{S}}} x_{q_1,..,q_N} \\
&= w_{\emptyset} \sum_{\substack{q_{i'} \neq p_{i'}, \\ i' \in \mathbb{N}}} x_{q_1,q_2,..,q_N} + w_{\{1\}} \sum_{\substack{q_{i'} \neq p_{i'}, \\ i' \in \mathbb{N}-\{1\}}} x_{p_1,q_2,..,q_N} + w_{\{2\}} \sum_{\substack{q_{i'} \neq p_{i'}, \\ i' \in \mathbb{N}-\{2\}}} x_{q_1,p_2,q_3,..,q_N} + \cdots \\
&\quad + w_{\{1,2\}} \sum_{\substack{q_{i'} \neq p_{i'}, \\ i' \in \mathbb{N}-\{1,2\}}} x_{p_1,p_2,q_3,...,q_N} + w_{\{1,3\}} \sum_{\substack{q_{i'} \neq p_{i'}, \\ i' \in \mathbb{N}-\{1,3\}}} x_{p_1,q_2,p_3,...,q_N} + \cdots + w_{\{1,...,N\}} x_{p_1,...,p_N} \\
&= w_{\emptyset} \sum_{q_1,...,q_N} x_{q_1,q_2,..,q_N} + (w_{\{1\}} - w_{\emptyset}) \sum_{q_2,...,q_N} x_{p_1,q_2,..,q_N} + \cdots + (w_{\{N\}} - w_{\emptyset}) \sum_{q_1,...,q_{N-1}} x_{q_1,..,q_{N-1},p_N} \\
&\quad + (w_{\{1,2\}} - w_{\emptyset} - w_{\{1\}} - w_{\{2\}}) \sum_{q_3,...,q_N} x_{p_1,p_2,q_3,...,q_N} + \cdots \\
&\quad + (w_{\{1,...,N\}} - w_{\emptyset} - w_{\{1\}} - \cdots - w_{\{N\}} - w_{\{1,2\}} - \cdots - w_{\{1,...,N-1\}}) x_{p_1,...,p_N} \\
&= \sum_{\mathbb{S} \subseteq \mathbb{N}} \left( w_{\mathbb{S}} - \sum_{\mathbb{U} \subset \mathbb{S}} w_{\mathbb{U}} \right) \sum_{\substack{q_i = p_i, i \in \mathbb{S} \\ q_i', i' \in \mathbb{N}\backslash\mathbb{S}}} x_{q_1,..,q_N}
\end{aligned}
\tag{15}
$$

We use $\hat{\mathbf{X}}_{\mathbb{A}} \in \mathbb{R}^{K_1 \times \cdots \times K_N}, \mathbb{A} \subseteq \mathbb{N}$ to represent the tensor obtained by applying the summation operation over the dimensions $\mathbb{A}$ of tensor $\mathbf{X}$, which is repeated over the dimensions $\mathbb{A}$ to match the

original shape. Note that $\hat{\mathbf{X}}_\emptyset = \mathbf{X}$. A single term in the above formula can be represented as follows

$$\left(w_{\mathbb{S}} - \sum_{\mathbb{U} \subset \mathbb{S}} w_{\mathbb{U}}\right) \sum_{\substack{q_i = p_i, i \in \mathbb{S} \\ q'_i, i' \in \mathbb{N} \backslash \mathbb{S}}} x_{q_1,..,q_N} = \bar{w}_{\mathbb{S}}[\hat{\mathbf{X}}_{\mathbb{N} \backslash \mathbb{S}}]_{p_1, p_2, ..., p_N}, \tag{16}$$

where $[\hat{\mathbf{X}}_{\mathbb{N} \backslash \mathbb{S}}]_{p_1, p_2, ..., p_N} \in \mathbb{R}$ denotes the $p_1, p_2, ..., p_N$-th element of $\hat{\mathbf{X}}_{\mathbb{N} \backslash \mathbb{S}}$, and $\bar{w}_{\mathbb{S}} = (w_{\mathbb{S}} - \sum_{\mathbb{U} \subset \mathbb{S}} w_{\mathbb{U}})$. Based on the above formula, we have $m_{p_1,...,p_N} = \sum_{\mathbb{S} \subseteq \mathbb{N}} \bar{w}_{\mathbb{S}}[\hat{\mathbf{X}}_{\mathbb{N} \backslash \mathbb{S}}]_{p_1, p_2, ..., p_N}$. Thus, $g_{\text{ETL}}(\mathbf{X})$ can be expressed as

$$g_{\text{ETL}}(\mathbf{X}) = \sigma\left(\sum_{\mathbb{S} \subseteq \mathbb{N}} \bar{w}_{\mathbb{S}} \hat{\mathbf{X}}_{\mathbb{N} \backslash \mathbb{S}}\right). \tag{17}$$

Then, we consider the output of SMPE framework. When $f_n = f_{\text{id}}, n = 1, ..., N$, equation equation 3 can be rewritten as

$$
\begin{aligned}
&g_{\text{SMPE}}(\mathbf{X}) \\
&= \sigma\left(\sum_{n=1}^{N} w_n^{\text{PE}}[f_n \circ f_{n-1} \circ \cdots \circ f_1(\mathbf{X})] + \sum_{\mathbb{S} \subseteq \mathbb{N} \backslash \{\emptyset\}} w_{\mathbb{S}}^{\text{GI}} \bar{\mathbf{X}}_{\mathbb{S}}\right) \\
&= \sigma\left(\sum_{n=1}^{N} w_n^{\text{PE}} \mathbf{X} + \sum_{\mathbb{P} \subseteq \mathbb{N} \backslash \{\emptyset\}} w_{\mathbb{P}}^{\text{GI}} \bar{\mathbf{X}}_{\mathbb{P}}\right) = \sigma\left[\left(\sum_{n=1}^{N} w_n^{\text{PE}}\right) \mathbf{X} + \sum_{\mathbb{P} \subseteq \mathbb{N} \backslash \{\emptyset\}} w_{\mathbb{P}}^{\text{GI}} \bar{\mathbf{X}}_{\mathbb{P}}\right].
\end{aligned}
\tag{18}
$$

Define $w_\emptyset^{\text{GI}} = \left(\sum_{n=1}^{N} w_n^{\text{PE}}\right)$, and we have

$$g_{\text{SMPE}}(\mathbf{X}) = \sigma\left(\sum_{\mathbb{P} \subseteq \mathbb{N}} w_{\mathbb{P}}^{\text{GI}} \bar{\mathbf{X}}_{\mathbb{P}}\right). \tag{19}$$

By replacing $\mathbb{P} = \mathbb{N} \backslash \mathbb{S}$, the output of the SMPE framework can be expressed as $g_{\text{SMPE}}(\mathbf{X}) = \sigma\left(\sum_{\mathbb{S} \subseteq \mathbb{N}} w_{\mathbb{N} \backslash \mathbb{S}}^{\text{GI}} \bar{\mathbf{X}}_{\mathbb{N} \backslash \mathbb{S}}\right)$. In comparison to equation 17, we can conclude that $g_{\text{SMPE}}(\mathbf{X}) = g_{\text{ETL}}(\mathbf{X})$ under one of the following conditions:

- $\bar{\mathbf{X}}$ is obtained by the summation operation, and $w_{\mathbb{N} \backslash \mathbb{S}}^{\text{GI}} = \bar{w}_{\mathbb{S}}$.

- $\bar{\mathbf{X}}$ is obtained by the mean operation, and $w_{\mathbb{N} \backslash \mathbb{S}}^{\text{GI}} = (\Pi_{n \in \mathbb{N} \backslash \mathbb{S}} K_n) \cdot \bar{w}_{\mathbb{S}}$.

This completes the proof for Proposition 3.2.

## C   MORE DETAILS ABOUT SMPE LAYER

In this section, we provide detailed implementations of the functions in the SMPE layer and the calculation of related computational complexities.

The architecture of SMPE layer SMPEL : $\mathbb{R}^{K_1 \times \cdots \times K_N \times D} \to \mathbb{R}^{K_1 \times \cdots \times K_N \times D}$ can be represented as follows:

$$\text{SMPEL}(\mathbf{X}) = \text{FC}_1(\sigma(\text{LN}(\mathbf{M}))), \tag{20}$$

$$\mathbf{M} = \text{FC}_2([\mathbf{O}_1, \mathbf{O}_2, ..., \mathbf{O}_N, \bar{\mathbf{X}}_{\mathbb{P}_1}, ..., \bar{\mathbf{X}}_{\mathbb{P}_{2^N-1}}]), \tag{21}$$

$$\mathbf{O}_0 = \mathbf{X}, \ \mathbf{O}_n = \text{PE1D}_n(\mathbf{O}_{n-1}), \ n = 1, 2, ..., N, \tag{22}$$

where we chose ReLU as the activation function $\sigma$ (Glorot et al., 2011). In addition to $\text{PE1D}_n, n = 1, ..., N$, the computational complexity of the SMPE layer mainly focuses on the fully connected layer $\text{FC}_2$ in equation 21, where its input and output dimensions are $K_1 \times \cdots \times K_N \times (N + 2^N - 1)D$ and $K_1 \times \cdots \times K_N \times D$. Since $\text{FC}_2$ operates at the last dimension, the first

$N$ dimensions of the input shape can be seen as the batch size. Thus, the computational complexity of this layer is $\mathcal{O}\left((\Pi_{n=1}^{N} K_n) \cdot (2^N + N + 1) \cdot D^2\right)$. $\sigma$ and LN are used for non-linearity and normalization, respectively. If the activation function and normalization layer are already included in $\text{PE1D}_n$, this step can be selectively omitted.

Since the input is a tensor instead of a matrix, before applying $\text{PE1D}_n : \mathbb{R}^{K_n \times D} \to \mathbb{R}^{K_n \times D}$, we performed the following operation:

$$\mathbf{X} \in \mathbb{R}^{B \times K_1 \times \cdots \times K_N \times D} \overset{\text{transpose}}{\longrightarrow} \mathbf{X} \in \mathbb{R}^{B \times K_1 \times \cdots \times K_{n-1} \times K_{n+1} \times \cdots \times K_N \times K_n \times D} \tag{23}$$

$$\overset{\text{reshape}}{\longrightarrow} \mathbf{X} \in \mathbb{R}^{(B K_1 \cdots K_{n-1} K_{n+1} \cdots K_N) \times K_n \times D}, \tag{24}$$

where $B$ denotes the batch size. After performing the above operation, the first dimension of $\mathbf{X}$ can be considered as the new batch dimension, and $\text{PE1D}_n$ can be applied to $\mathbf{X}$. Subsequently, $\mathbf{X}$ is restored to its original tensor dimensions through the reverse operation of equation 23 and equation 24.

Then, we consider the two specific implementations of $\text{PE1D}_n$. It is worth noting that we employ both of these structures in our experiments for illustration purposes. In practice, any one-dimensional equivariant network layer can be applied within the SMPE framework.

## C.1 SMPE-DS Layer

SMPE-DS layer applys the 1D equivariant layers in Zaheer et al. (2017) as the functions $\text{PE1D}_n$, $n = 1, ..., N$. We further employ LN as the normalization function, and the layer $\text{PE1D}_n$ can be represented as follows:

$$\text{PE1D}_n(\mathbf{X}) = \sigma\left(\text{LN}(\mathbf{1}\boldsymbol{\beta}^T + [\mathbf{X} - \mathbf{1}\text{pool}(\mathbf{X})]\boldsymbol{\Gamma})\right), \tag{25}$$

where $\mathbf{X} \in \mathbb{R}^{K_n \times D}$ is the input, $\mathbf{1} \in \mathbb{R}^{K_n \times 1}$ represents a column vector of all ones, $\boldsymbol{\Gamma} \in \mathbb{R}^{D \times D}$ and $\boldsymbol{\beta} \in \mathbb{R}^{D \times 1}$ are trainable parameters. $\text{pool} : \mathbb{R}^{K_n \times D} \to \mathbb{R}^{1 \times D}$ is the pool function, which can be $\text{maxpool}$, $\text{meanpool}$, and $\text{sumpool}$. Without loss of generality, in our experimental section, we used the $\text{meanpool}$41 and choosed ReLU as the activation function $\boldsymbol{\sigma}$. According to equation 24 and equation 25, computational complexity of $\text{PE1D}_n$ is $\mathcal{O}\left((\Pi_{n=1}^{N} K_n)D^2\right)$. For all of $\text{PE1D}_n, n = 1, ..., N$, the complexity order is $\mathcal{O}\left((\Pi_{n=1}^{N} K_n)N \cdot D^2\right)$. The overall computational complexity orders for equation 21 and equation 22 remain at $\mathcal{O}\left((\Pi_{n=1}^{N} K_n) \cdot (2^N + N + 1) \cdot D^2\right)$.

## C.2 SMPE-TF Layer

SMPE-TF layer applys the transformer encoder layer (without positional encoding) (Vaswani et al., 2017) as the functions $\text{PE1D}_n$, $n = 1, ..., N$, which is similar to the 1D permutation equivariant layer in Lee et al. (2019). According to equation 24, the dimensions of $\mathbf{Q}_n$, $\mathbf{K}_n$, and $\mathbf{V}_n$ in the layer are both $\Pi_{j=1, j \neq n}^{N} K_j \times K_n \times D$, where the first dimension can be seen as the batch size. Thus the computational complexity of $\text{PE1D}_n$ is $\mathcal{O}(\Pi_{j=1, j \neq n}^{N} K_j \cdot K_n^2 \cdot D)$ (Vaswani et al., 2017). For all of $\text{PE1D}_n, n = 1, ..., N$, the complexity order is $\mathcal{O}\left((\Pi_{n=1}^{N} K_n) \cdot (\sum_{n=1}^{N} K_n) \cdot D\right)$. Thus, the overall computational complexity orders for equation 21 and equation 22 is $\mathcal{O}\left((\Pi_{n=1}^{N} K_n) \cdot [(2^N + N - 1)D + \sum_{n=1}^{N} K_n] \cdot D\right)$.

## C.3 Complexity Comparison

We analyze the computational complexity and the parameters of the proposed framework. We also compare the exchangeable tensor layer (denoted by ETL) in Hartford et al. (2018), and the results are shown in Table 4. As shown in the table, the number of parameters displayed is independent of the number of objects $K_1, ..., K_N$ in each dimension. This implies that, similar to ETL, the proposed SMPE framework is suitable for inputs with varying numbers of objects. On the other hand, for a given number of dimensions $N$, both SMPEL$-$DS and ETL exhibit computational complexity that is linearly related to the total number of objects $K_{\text{all}} = \Pi_{n=1}^{N} K_n$. Although SMPEL$-$TF employs the high-complexity self-attention mechanism, the dimension-wise processing ensures that the computational complexity is lower than second-order terms of $K_{\text{all}}$, i.e.,

$(\Pi_{n=1}^{N} K_n) \sum_{n=1}^{N} K_n \le (\Pi_{n=1}^{N} K_n)^2$. As $N$ grows larger or as $K_1, ..., K_N$ become closer, the difference between the expressions on both sides of the inequality increases.

Table 4: Comparison of computational complexity and parameters.

| Layer Type | Complexity | Parameters |
|---|---|---|
| SMPEL−DS | $\mathcal{O}\left((\Pi_{n=1}^{N} K_n) \cdot (2^N + N + 1) \cdot D^2\right)$ | $\mathcal{O}\left((2^N - 1 + N)D^2\right)$ |
| SMPEL−TF | $\mathcal{O}\left((\Pi_{n=1}^{N} K_n) \cdot [(2^N + N - 1)D + \sum_{n=1}^{N} K_n] \cdot D\right)$ | $\mathcal{O}\left((2^N - 1 + N)D^2\right)$ |
| ETL (Hartford et al., 2018) | $\mathcal{O}\left((\Pi_{n=1}^{N} K_n) \cdot 2^N \cdot D^2\right)$ | $\mathcal{O}(2^N D^2)$ |

# D  TASKS DESCRIPTION

## D.1  PSEUDO INVERSE COMPUTATION

We consider the computation of the transposed pseudo inverse $(\boldsymbol{X}^{\dagger})^T = (\boldsymbol{X}^H (\boldsymbol{X}\boldsymbol{X}^H)^{-1})^T \in \mathbb{C}^{K \times N}$ of the input matrix $\boldsymbol{X} \in \mathbb{C}^{K \times N}, K < N$. The mapping from $\boldsymbol{X}$ to $(\boldsymbol{X}^{\dagger})^T$ is 2D equivariant. The networks are trained to compute $(\boldsymbol{X}^{\dagger})^T$ based on $\boldsymbol{X}$.

## D.2  TRANSMISSION DESIGN FOR MIMO SYSTEM

We consider the transmission design problem in MIMO systems, specifically focusing on the precoding design for downlink transmission. Precoding is a crucial and widely investigated technique in wireless communication systems (Albreem et al., 2021). Consider a MIMO downlink system where an $N$-antenna BS transmits the signal to $K$ single-antenna UE. The channel between BS and the $k$-th UE is denoted as $\boldsymbol{h}_k \in \mathbb{C}^{N \times 1}$. The channel matrix $\boldsymbol{H} = [\boldsymbol{h}_1, \boldsymbol{h}_2..., \boldsymbol{h}_K]^T \in \mathbb{C}^{K \times N}$ is assumed to be available at the BS. Building upon $\boldsymbol{H}$ and the noise variance $\sigma^2$, the precoding matrix $\boldsymbol{W} = [\boldsymbol{w}_1, \boldsymbol{w}_2..., \boldsymbol{w}_K]^T \in \mathbb{C}^{K \times N}$ is designed to enhance the performance of downlink transmission. The most widely adopted design criterion for $\boldsymbol{W}$ is maximizing the sum rate under the power constraint $P_{\mathrm{T}}$, i.e.,

$$\boldsymbol{W}^{\star} = \arg\max R_{\mathrm{M}}, \quad \text{s.t. } \mathrm{Tr}(\boldsymbol{W}\boldsymbol{W}^H) = P_{\mathrm{T}}, \tag{26}$$

where $R_{\mathrm{M}}$ is the sum rate for MIMO transmission given by

$$R_{\mathrm{M}} = \sum_{k=1}^{K} \log_2 \left( 1 + \frac{|\boldsymbol{h}_k^H \boldsymbol{w}_k|^2}{\sigma^2 + \sum_{i \ne k} |\boldsymbol{h}_k^H \boldsymbol{w}_i|} \right). \tag{27}$$

Due to the non-convex nature of problem equation 26, solving it is challenging. We model the solution of the above problem as $\boldsymbol{W}^{\star} = f(\boldsymbol{H})$ for a given $\sigma^2$, and we have $f(\pi^{\{1,2\}} \circ \boldsymbol{H}) = \pi^{\{1,2\}} \circ f(\boldsymbol{H}) = \pi^{\{1,2\}} \circ \boldsymbol{W}^{\star}$, which means such a function can be seen as a 2D equivariant function. The networks are trained to obtain $\boldsymbol{W}^{\star}$ based on $\boldsymbol{H}$ and SNR $\frac{P_T}{\sigma^2}$.

## D.3  TRANSMISSION DESIGN FOR CELL-FREE MIMO SYSTEM

Similar to subsection D.2, we further consider the precoding design for cell-free MIMO systems (Feng et al., 2022). There is a CPU controlling $M$ distributed APs to support $K$ single-antenna UEs, where each AP is equipped with $N$ antennas. The channel between $K$ UEs and the $m$-th AP is denoted as $\boldsymbol{H}_m = [\boldsymbol{h}_{m,1}, \boldsymbol{h}_{m,2}..., \boldsymbol{h}_{m,K}]^T \in \mathbb{C}^{K \times N}$, and the corresponding precoding matrix is $\boldsymbol{W}_m = [\boldsymbol{w}_{m,1}, \boldsymbol{w}_{m,2}..., \boldsymbol{w}_{m,K}]^T \in \mathbb{C}^{K \times N}$. By stacking the channel and precoding matrices belonging to all APs, we have $\mathsf{H} \in \mathbb{C}^{M \times K \times N}$ and $\mathsf{W} \in \mathbb{C}^{M \times K \times N}$. Then, the optimization problem of $\mathsf{W}$ is given by

$$\boldsymbol{W}^{\star} = \arg\max R_{\mathrm{CFM}}, \quad \text{s.t. } \mathrm{Tr}(\boldsymbol{W}_m \boldsymbol{W}_m^H) = P_{\mathrm{T}}, \ m = 1, ..., M, \tag{28}$$

where $\boldsymbol{h}_k = [\boldsymbol{h}_{1,k}^T, \boldsymbol{h}_{2,k}^T, ..., \boldsymbol{h}_{M,k}^T]^T \in \mathbb{C}^{MN \times 1}$ and $\boldsymbol{w}_k = [\boldsymbol{w}_{1,k}^T, \boldsymbol{w}_{2,k}^T, ..., \boldsymbol{w}_{M,k}^T]^T \in \mathbb{C}^{MN \times 1}$. $R_{\mathrm{CFM}}$ is the sum rate for cell-free MIMO transmission expressed as

$$R_{\mathrm{CFM}} = \sum_{k=1}^{K} \log_2 \left( 1 + \frac{|\boldsymbol{h}_k^H \boldsymbol{w}_k|^2}{\sigma^2 + \sum_{i \ne k} |\boldsymbol{h}_k^H \boldsymbol{w}_i|} \right). \tag{29}$$

We model the solution to problem equation 28 as $\mathbf{W}^\star = f(\mathbf{H})$ for a given $\sigma^2$, and we have $f(\pi^3 \circ \mathbf{H}) = \pi^3 \circ f(\mathbf{H}) = \pi^3 \circ \mathbf{W}^\star$, which means such a function can be seen as a 3D permutation equivariant function. The networks are trained to obtain $\mathbf{W}^\star$ based on $\mathbf{H}$ and SNR.

# E EXPERIMENTAL SETUP

## E.1 PSEUDO INVERSE COMPUTATION

We concatenate the real and imaginary parts of $\boldsymbol{X}$ at the newly appended last dimension to obtain $\mathbf{X} \in \mathbb{R}^{K \times N \times 2}$, and the reverse procedure is employed at output $\mathbf{Y} \in \mathbb{R}^{K \times N \times 2}$ to obtain the result $\boldsymbol{Y} \in \mathbb{C}^{K \times N}$. Without loss of generality, $\boldsymbol{X}$ is sampled from random matrices whose elements are independently identical distributed (i.i.d.) with complex Gaussian distribution $\mathcal{CN}(0,1)$. We use MAE and mean squared error (MSE) as the loss functions. It is easy to prove that the transposed pseudo inverse is a 2D permutation equivariant function for $\mathbf{X}$, where the feature dimension is $D_X = D_Y = 2$. The input is flattened to $\boldsymbol{X} \in \mathbb{R}^{KN \times 2}$ for `SetTrans1D`. The evaluation metrics of this experiment are chosen to be the MAE and MSE losses between $\boldsymbol{Y}$ and $(\boldsymbol{X}^\dagger)^T$, which are separately computed and summed for the real and imaginary parts, i.e.,

$$\text{MAE} = (\|\Re(\boldsymbol{Y}) - \Re(\boldsymbol{X}^\dagger)^T\|_1 + \|\Im(\boldsymbol{Y}) - \Im(\boldsymbol{X}^\dagger)^T\|_1)/KN, \tag{30}$$

$$\text{MSE} = (\|\Re(\boldsymbol{Y}) - \Re(\boldsymbol{X}^\dagger)^T\|_F^2 + \|\Im(\boldsymbol{Y}) - \Im(\boldsymbol{X}^\dagger)^T\|_F^2)/KN. \tag{31}$$

We train networks separately for each evaluation metric, and the loss function during training is the average of the evaluation metric from the data in each batch.

## E.2 TRANSMISSION DESIGN FOR MIMO SYSTEM

With the fixed $P_\text{T} = 1$, we train these networks on seven different SNRs, i.e., $\frac{P_\text{T}}{\sigma^2} = \{0, 5, 10, 15, 20, 25, 30\}$. Following the same procedure as described in subsection E.1, we concatenate the real parts, imaginary parts, and $\sigma^2$ at the newly expanded last dimension. Consequently, the shapes of the input and output are $\widetilde{\mathbf{H}} \in \mathbb{R}^{K \times N \times 3}$ and $\widetilde{\mathbf{W}} \in \mathbb{R}^{K \times N \times 2}$, respectively. The input of `SetTrans1D` is flattened to $\widetilde{\boldsymbol{H}} \in \mathbb{R}^{KN \times 3}$. The dataset of $\boldsymbol{H}$ is sampled from the channels generated by the Rayleigh channel model (Telatar, 1999), which means its elements follow i.i.d. Gaussian distribution $\mathcal{CN}(0,1)$. To ensure the result $\boldsymbol{W}$ reshaped from $\widetilde{\mathbf{W}}$ satisfies the power constraint in equation 26, we use $\sigma_\text{out}(\boldsymbol{W}) = \sqrt{P_\text{T}/\text{Tr}(\boldsymbol{W}\boldsymbol{W}^H)} \cdot \boldsymbol{W}$ as the output layer of networks. The evaluation metric $\bar{R}_\text{M}$ for this experiment is defined as the average of $R_\text{M}$ across seven SNR values $\{0, 5, 10, 15, 20, 25, 30\}$, where the expression of $R_\text{M}$ is given by equation 27. The loss function during training is the average of the negative $\bar{R}_\text{M}$ computed from the data in each batch.

## E.3 TRANSMISSION DESIGN FOR CELL-FREE MIMO SYSTEM

Following the similar procedure in subsection E.1, the dimensions of the input and output are $\widetilde{\mathbf{H}} \in \mathbb{R}^{M \times K \times N \times 3}$ and $\widetilde{\mathbf{W}} \in \mathbb{R}^{M \times K \times N \times 2}$, respectively. The input for 2D networks is flattened to $\widetilde{\mathbf{H}} \in \mathbb{R}^{K \times MN \times 3}$. To ensure that $\{\boldsymbol{W}_m\}_{m=1}^M$ reshaped from $\widetilde{\mathbf{W}}$ satisfy the power constraint in equation 28, we use $\sigma_\text{out}(\boldsymbol{W}_m) = \sqrt{P_\text{T}/\text{Tr}(\boldsymbol{W}_m\boldsymbol{W}_m^H)} \cdot \boldsymbol{W}_m$ as the output layer of networks. The evaluation metric $\bar{R}_\text{CFM}$ is defined as the average of $R_\text{CFM}$ across seven SNR values $\{0, 5, 10, 15, 20, 25, 30\}$, where the expression of $R_\text{CFM}$ is given by equation 29. The the loss function during training is the average of the negative $\bar{R}_\text{CFM}$ from the data in each batch. The dataset of $\mathbf{H}$ is sampled from the channels generated by the QUAsi Deterministic RadIo channel GenerAtor (QuaDRiGa) (Jaeckel et al., 2014), which is used for generating realistic radio channel impulse responses for system-level simulations of mobile radio networks.

## E.4 COMMON SETUP

In this subsection, we provide some common experimental configurations and hyperparameters.

- **Pooling functions:** Without loss of generality, all the pooling operations are set to be the mean function.

- **The number of layers:** Except for ETN2D and ETN3D, which consist of eight layers, all the involved networks comprise four layers.
- **Feature dimension $D$:** For all networks, the feature dimension $D$ in the hidden layers is set to 256.
- **Training strategy:** We perform batch iterations for $8 \times 10^4$ times with a batch size of 512 and reduce the learning rate by half after $4 \times 10^4$ iterations. This strategy for the three tasks is the same. For the design task of MIMO and cell-free MIMO system transmission, we randomly assign SNR values to each training and testing sample.
- **Dataset size:** For each task, we use $2 \times 10^5$ training samples and $5 \times 10^3$ testing samples.
- **Activation & normalization:** To accelerate convergence and enhance the stability of the training process, we applied the ReLU activation and LN to ETN2D and ETN3D.
- **Parameter initialization:** With the exception of networks SMPEN2D$-$TF and SMPEN3D$-$TF in the task of subsections 4.3, which were initialized using the default method, we employed Kaiming initialization (He et al., 2015) for the weight parameters of the linear layers in networks used for other tasks.
- **Monte-Carlo simulation:** Each experimental result is the average of three valid experiments.

## F  ADDITIONAL EXPERIMENTAL RESULT

### F.1  ABLATION STUDY

To validate the superiority of $g_{\text{serial}}$ (represented by ParaN2D$-$TF) over $g_{\text{para}}$ (represented by SerialN2D$-$TF) in subsection 3.2, as well as the superiority of $g_{\text{SMPE}}$ over $g_{\text{serial}}$, we compared the performance of these three methods on task in subsection 4.2. It can be observed that in most scenarios, $g_{\text{serial}}$ outperforms $g_{\text{para}}$, and $g_{\text{SMPE}}$ outperforms $g_{\text{serial}}$.

Table 5: Experiment results of ablation study

| | Test Setting | $\mathcal{A}$: $N=7, K=7$ | | | $\mathcal{B}$: $N=8, K=7$ | | | $\mathcal{C}$: $N=8, K=8$ | | |
|---|---|---|---|---|---|---|---|---|---|---|
| | **Train Setting** | $\mathcal{A}$ | $\mathcal{B}$ | $\mathcal{C}$ | $\mathcal{A}$ | $\mathcal{B}$ | $\mathcal{C}$ | $\mathcal{A}$ | $\mathcal{B}$ | $\mathcal{C}$ |
| $\bar{R}_{\text{M}}(\uparrow)$ | ParaN2D$-$TF | 20.49 | 20.39 | 20.33 | 21.72 | 21.87 | 21.62 | 21.99 | 21.61 | 21.97 |
| | SerialN2D$-$TF | 20.75 | 21.48 | 19.08 | 21.91 | 22.94 | 20.31 | 19.26 | 20.18 | 24.33 |
| | SMPEN2D$-$TF | **22.21** | 22.70 | 20.69 | 23.78 | **24.71** | 22.30 | 21.80 | 23.17 | **24.44** |

### F.2  RUNNING TIME COMPARISON

Table 6 provides a comparison of the runtime for each network across different tasks. The runtime refers to the average time taken for a batch during inference on GPU, with a batch size of 1000. In

Table 6: Running time (ms) comparison.

| | $1\mathcal{A}$ | $1\mathcal{B}$ | $1\mathcal{C}$ | $2\mathcal{A}$ | $2\mathcal{B}$ | $2\mathcal{C}$ | $3\mathcal{A}$ | $3\mathcal{B}$ | $3\mathcal{C}$ |
|---|---|---|---|---|---|---|---|---|---|
| SetTrans1D | 61.7 | 67.5 | 72.1 | 16.4 | 18.6 | 24.3 | | | |
| CITransNet2D | 87.9 | 106.0 | 119.1 | 48.2 | 53.8 | 64.4 | 34.8 | 34.9 | 63.2 |
| ETN2D | 61.9 | 69.2 | 75.8 | 18.7 | 21.9 | 28.4 | 13.8 | 13.8 | 27.1 |
| SMPEN2D$-$DS | 65.2 | 75.6 | 80.4 | 21.3 | 24.6 | 31.4 | 12.8 | 12.9 | 24.7 |
| SMPEN2D$-$TF | 87.8 | 109.1 | 129.0 | 51.2 | 59.9 | 69.1 | 35.0 | 35.1 | 63.4 |
| ETN3D | | | | | | | 25.6 | 25.7 | 46.9 |
| SMPEN3D$-$DS | | | | | | | 23.1 | 23.2 | 42.5 |
| SMPEN3D$-$TF | | | | | | | 55.4 | 55.5 | 99.3 |

the first row, '1', '2' and '3' denote experiments in subsections 4.1, 4.2, and 4.3, respectively. $\mathcal{A}$, $\mathcal{B}$, and $\mathcal{C}$ represent the scenario configurations for each task during inference. It can be observed that the runtime of $\mathtt{SMPEN-DS}$ is comparable to that of $\mathtt{ETN}$, and the runtime of $\mathtt{SMPEN2D-TF}$ is similar to that of $\mathtt{CITransNet2D}$.

## F.3 CONTEXTUAL AUCTION DESIGN

Since $\mathtt{CITransNet2D}$ (Duan et al., 2022) and $\mathtt{SMPEN2D-TF}$ are 2D equivariant networks both based on the transformer, we compare them on the task of contextual auction design in Duan et al. (2022). Based on the avaliable contexts, contextual auction design targets to find the optimal allocation and payment rules to maximize the expected revenue ($rev$) while maintaining a near-zero level of dominant strategy incentive compatibility ($rgt$). The contextual auction contains $N$ bidders and $M$ items, and their contexts are represented by $\boldsymbol{X} = [\boldsymbol{x}_1, \boldsymbol{x}_2, ..., \boldsymbol{x}_N]^T \in \mathbb{R}^{N \times d_x}$ and $\boldsymbol{Y} = [\boldsymbol{y}_1, \boldsymbol{y}_2, ..., \boldsymbol{y}_M]^T \in \mathbb{R}^{M \times d_y}$. The optimal allocation rule matrix $\boldsymbol{G} \in \mathbb{R}^{N \times M}$ and the payment rule vector $\boldsymbol{p} = [p_1, ..., p_N]^T \in \mathbb{R}^{N \times 1}$ are designed based on $\boldsymbol{X}$, $\boldsymbol{Y}$, and the bidding profile $\boldsymbol{B} \in \mathbb{R}^{N \times M}$. The input tensor $\mathbf{X} \in \mathbb{R}^{N \times M \times (d_x + d_y + 1)}$ can be obtained by repeating $\boldsymbol{X}$ and $\boldsymbol{Y}$ and concatenating them with $\boldsymbol{B}$. The dimension of output is set to $\mathbf{Y} \in \mathbb{R}^{N \times M \times 3}$, where $\mathbf{G}$ and $\boldsymbol{p}$ can be obtained from $\mathbf{Y}_{[:,:,1:2]}$ and $\mathbf{Y}_{[:,:,3]}$. The definition of this problem, the output layer of the network, and the dataset are intricate, whose details can be found in (Duan et al., 2022). It can be verified that the mapping from $\mathbf{X}$ to $\mathbf{Y}$ is 2D equivariant.

Table 7: Experiment results about contextual auction design.

| Method | $N=2, M=5$ $\|\mathbb{X}\| = \|\mathbb{Y}\| = 10$ | | $N=3, M=10$ $\|\mathbb{X}\| = \|\mathbb{Y}\| = 10$ | | $N=2, M=5$ $\mathbb{X}, \mathbb{Y} \in \mathbb{R}^{10}$ | | $N=3, M=10$ $\mathbb{X}, \mathbb{Y} \in \mathbb{R}^{10}$ | |
|---|---|---|---|---|---|---|---|---|
| | $rev$ | $rgt$ | $rev$ | $rgt$ | $rev$ | $rgt$ | $rev$ | $rgt$ |
| $\mathtt{CITransNet2D}$ | 2.901 | $< 0.001$ | 6.858 | $< 0.001$ | 1.126 | $< 0.001$ | 2.931 | $< 0.001$ |
| $\mathtt{SMPEN2D-TF}$ | **2.955** | $< 0.001$ | **6.880** | $< 0.001$ | **1.182** | $< 0.001$ | **2.986** | $< 0.001$ |

Table 7 presents a comparison of $rev$ and $rgt$ under different configurations of $N$, $M$, $\mathbb{X}$, and $\mathbb{Y}$. The results show that $\mathtt{SMPEN2D}$ outperforms $\mathtt{CITransNet2D}$ in terms of $rev$ while keeping $rgt$ sufficiently low.

## F.4 TRANSMISSION DESIGN FOR MIMO SYSTEM

For readability purposes, we only provided the performance of $\bar{R}_{\mathrm{M}}$ in Table 2. We present the performance of $R_{\mathrm{M}}$ under seven SNR values in figure 4 to provide a more detailed performance comparison.

## F.5 TRANSMISSION DESIGN FOR CELL-FREE MIMO SYSTEM

Table 8: Transmission performance $\bar{R}_{\mathrm{CFM}}$ (the higher the better) comparisons for cell-free MIMO system.

| | Test Setting | $\mathcal{A}$: $M=3, K=6, N=2$ | | | $\mathcal{B}$: $M=2, K=6, N=3$ | | | $\mathcal{C}$: $M=4, K=8, N=2$ | | |
|---|---|---|---|---|---|---|---|---|---|---|
| | Train Setting | $\mathcal{A}$ | $\mathcal{B}$ | $\mathcal{C}$ | $\mathcal{A}$ | $\mathcal{B}$ | $\mathcal{C}$ | $\mathcal{A}$ | $\mathcal{B}$ | $\mathcal{C}$ |
| | $\mathtt{CITransNet2D}$ | 21.81 | 21.03 | 24.52 | 21.33 | 21.02 | 22.89 | 20.49 | 19.85 | 27.64 |
| | $\mathtt{ETN2D}$ | 22.33 | 21.28 | 18.99 | 21.59 | 21.23 | 18.68 | 18.06 | 17.19 | 27.87 |
| $\bar{R}_{\mathbf{CFM}}(\uparrow)$ | $\mathtt{ETN3D}$ | 24.11 | 14.02 | 18.88 | 13.75 | 23.57 | 14.43 | 15.36 | 12.03 | 27.61 |
| | $\mathtt{SMPEN2D-DS}$ | 22.91 | 21.80 | 19.45 | 22.50 | 22.02 | 19.20 | 18.31 | 17.46 | 29.26 |
| | $\mathtt{SMPEN3D-DS}$ | 25.02 | 13.13 | 18.66 | 13.61 | 24.49 | 14.05 | 13.98 | 10.51 | 30.40 |
| | $\mathtt{SMPEN2D-TF}$ | 24.27 | 23.08 | 23.33 | 23.84 | 23.30 | 23.04 | 21.15 | 20.13 | 32.74 |
| | $\mathtt{SMPEN3D-TF}$ | **26.61** | 12.54 | 21.66 | 15.37 | **25.53** | 15.56 | 19.48 | 11.00 | **34.37** |

The experimental results based on the Rayleigh channel model are shown in Table 8. Similar to the MIMO system, we also list the performance of $R_{\text{CFM}}$ under seven SNR values in figure 5.

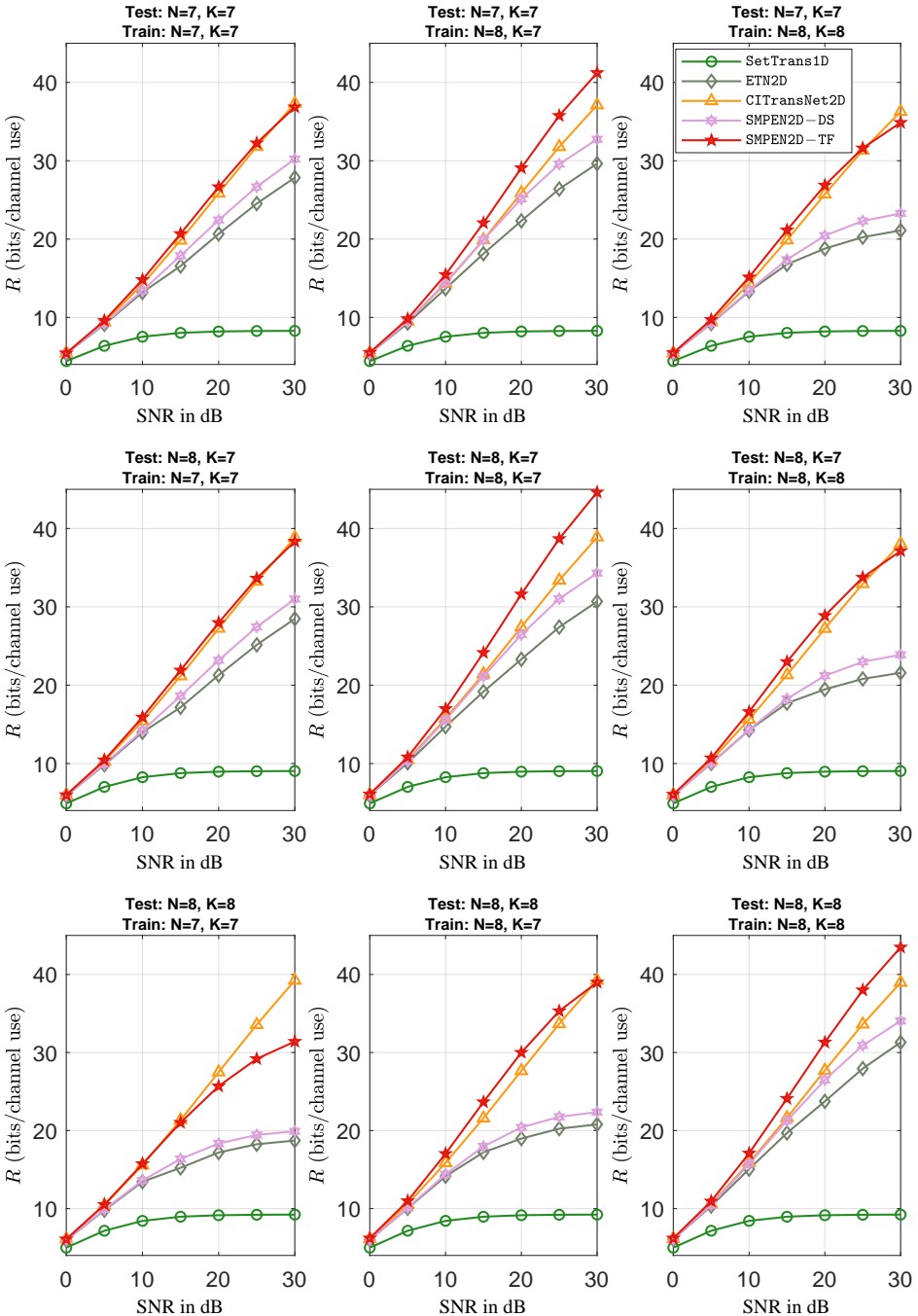

Figure 4: $R_{\text{M}}$ vs SNR, MIMO transmission.

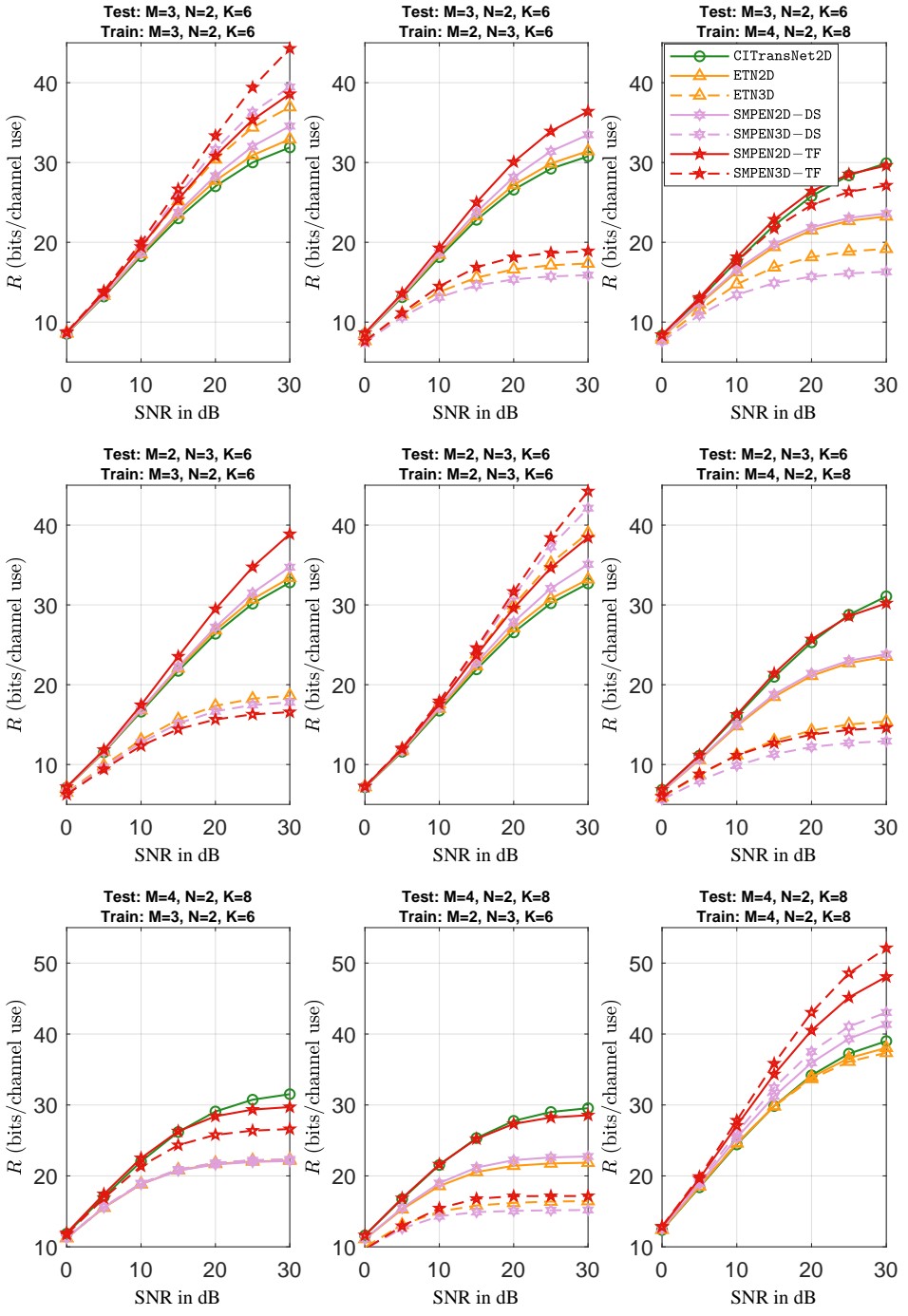

Figure 5: $R_{\mathrm{CFM}}$ vs SNR, Cell-free MIMO transmission.

