# OpenReview forum: "SMPE: A Framework for Multi-Dimensional Permutation Equivariance"
_ICLR.cc/2024/Conference — Submitted to ICLR 2024_

### Official Review · Reviewer_y2bA · 2023-10-19

**Soundness:** 3 good
**Presentation:** 2 fair
**Contribution:** 3 good
**Rating:** 6
**Confidence:** 2

**Summary:**

This paper proposes a multi-dimensional permutation equivariance framework named SMPE to pave the way for the design of multi-dimensional permutation equivariant networks.

**Strengths:**

- This work provides the first exact algebra-based definition of multi-dimensional permutation equivariance.

- The experiments are extensive, including contextual auction design, pseudo inverse computation, and typical wireless communication tasks.

- The research can benefit many real applications including point cloud analysis, graph analysis, pseudo inverse computation, typical wireless communication, etc.

- The proposed technique seems sound, and its effectiveness is verified by experiments.

**Weaknesses:**

- The running time of all the methods could be compared to show the efficiency of the proposed method.

- The hyperparameter settings of the model should be listed in the text.

- Two related works [1,2] about graph permutation invariance are missing in the discussion.

- The source codes and the datasets could be provided to facilitate the reproducibility of this work.

Refs:

[1] [arXiv 2019] PiNet: A Permutation Invariant Graph Neural Network for Graph Classification

[2] [TKDE 2022] Graph Substructure Assembling Network with Soft Sequence and Context Attention

**Questions:**

Please respond to the weaknesses listed in the previous text box.

**Details Of Ethics Concerns:**

I have not identified any ethical concerns about this work.

---

> ### Author Response · Authors · 2023-11-20
> **Comparison of running times to show efficiency**
>
> Many thanks for your careful reviews and constructive comments.
>
> The running times of all the methods have been compared in Appendix F.2. Specifically, the following table provides a comparison of the runtime for each network across different tasks. The runtime refers to the average time (ms) taken for a batch during GPU inference, with a batch size of 1000. In the first row, '1',  '2' and '3' denote experiments in Sections 4.1, 4.2, and 4.3, respectively. ${\mathcal{A}}$, ${\mathcal{B}}$, and ${\mathcal{C}}$ represent the scenario configurations for each task during inference. It can be observed that the running time of $\mathtt{SMPEN-DS}$ is comparable to that of $\mathtt{ETN}$, and the running time of $\mathtt{SMPEN2D-TF}$ is similar to that of $\mathtt{CITransNet2D}$.
>
> |                                | 1$\\mathcal{A}$ | 1$\\mathcal{B}$ | 1$\\mathcal{C}$ | 2$\\mathcal{A}$ | 2$\\mathcal{B}$ | 2$\\mathcal{C}$ | 3$\\mathcal{A}$ | 3$\\mathcal{B}$ | 3$\\mathcal{C}$ |
> | :----------------------------: | :---------------: | :---------------: | :---------------: | :---------------: | :---------------: | :---------------: | :---------------: | :---------------: | :---------------: |
> | $\\mathtt{SetTrans1D}$       | 61\.7             | 67\.5             | 72\.1             | 16\.4             | 18\.6             | 24\.3             |                   |                   |                   |
> | $\\mathtt{CITransNet2D}$     | 87\.9             | 106\.0            | 119\.1            | 48\.2             | 53\.8             | 64\.4             | 34\.8             | 34\.9             | 63\.2             |
> | $\\mathtt{ETN2D}$            | 61\.9             | 69\.2             | 75\.8             | 18\.7             | 21\.9             | 28\.4             | 13\.8             | 13\.8             | 27\.1             |
> | $\\mathtt{SMPEN2D\\!-\\!DS}$ | 65\.2             | 75\.6             | 80\.4             | 21\.3             | 24\.6             | 31\.4             | 12\.8             | 12\.9             | 24\.7             |
> | $\\mathtt{SMPEN2D\\!-\\!TF}$ | 87\.8             | 109\.1            | 129\.0            | 51\.2             | 59\.9             | 69\.1             | 35\.0             | 35\.1             | 63\.4             |
> | $\\mathtt{ETN3D}$            |                   |                   |                   |                   |                   |                   | 25\.6             | 25\.7             | 46\.9             |
> | $\\mathtt{SMPEN3D\\!-\\!DS}$ |                   |                   |                   |                   |                   |                   | 23\.1             | 23\.2             | 42\.5             |
> | $\\mathtt{SMPEN3D\\!-\\!TF}$ |                   |                   |                   |          |                   |                    | 55\.4             | 55\.5             | 99\.3             |

---

> ### Author Response · Authors · 2023-11-20
> **Listing of network hyperparameter configurations**
>
> The hyperparameters used in our experiments are listed in Appendix E.4
> of the revised paper. For example
>
> -   Pooling functions: Without loss of generality, all the pooling
>     operations are set to be the mean function.
>
> -   Training strategy: We perform batch iterations for $8\times 10^{4}$
>     times with a batch size of 512, and reduce the learning rate by half
>     after $4\times 10^{4}$ iterations. This strategy for three tasks is
>     the same. For the design task of MIMO and cell-free MIMO system
>     transmission, we randomly assign SNR values to each training and
>     testing sample.
>
> -   Activation & normalization: To accelerate convergence and enhance
>     the stability of the training process, we applied the ReLU
>     activation and LN to $\mathtt{ETN2D}$ and $\mathtt{ETN3D}$.
>
> -   Monte-Carlo simulation: Each experimental result is the average of
>     three valid experiments.

---

> ### Author Response · Authors · 2023-11-20
> **Discussion about the two related works [1][2]**
>
> In \[1\], a **one-dimensional** invariant network was constructed based
> on the proposed variable node attention pooling mechanism, which
> effectively addressed **graph classification problems**. Due to its
> permutation invariance, it is applicable to graphs of different sizes.
>
> The work in \[2\] introduces the Substructure Assembling Network (SAN),
> a graph neural network model that effectively learns interpretable and
> discriminative graph representations for **classification**. It
> overcomes challenges by hierarchically assembling substructures using
> the Substructure Assembling Unit (SAU) and enhancing performance with
> Soft Sequence with Context Attention (SSCA).
>
> The goal of these two works is to model **one-dimensional** invariant
> mappings, which is relevant to our work but not the primary focus of our
> investigation. We have incorporated these two works into the 'Related
> Work' section and provided additional discussions.
>
> \[1\] Meltzer P, Mallea M D G, Bentley P J. Pinet: A permutation
> invariant graph neural network for graph classification\[J\]. arXiv
> preprint arXiv:1905.03046, 2019.
>
> \[2\] Yang Y, Guan Z, Zhao W, et al. Graph substructure assembling
> network with soft sequence and context attention\[J\]. IEEE Transactions
> on Knowledge and Data Engineering, 2022, 35(5): 4894-4907.

---

> ### Author Response · Authors · 2023-11-20
> **The source codes and the datasets for reproducibility**
>
> The source code and associated datasets will be provided soon, and we
> are currently making efforts to organize the code and add comments to
> improve reproducibility.

---

### Official Review · Reviewer_QBbk · 2023-10-24

**Soundness:** 3 good
**Presentation:** 3 good
**Contribution:** 2 fair
**Rating:** 6
**Confidence:** 4

**Summary:**

The authors propose a method for obtaining neural networks with higher-order permutation equivariances. This concerns tensors where multiple dimensions are equivariant. The main approach uses a composition of first order permutation equivariant functions. Features are pooled from all subsets of dimensions to facilitate exchange of information between the different dimensions.

**Strengths:**

- Straightforward and easy to understand method, well-written
- I believe the method to be novel
- While there is existing work handling this subject, I believe that this is a useful *practical* contribution to the literature.
- Interesting applications used in experiments, solid results

**Weaknesses:**

- I believe that this topic has been studied before as part of more general treatments of permutation equivariance, such as in [1] under the name of higher-order permutations. As such, the techniques used themselves are fairly standard and novelty therefore limited, even if their exact combination is new.
- The experiments could be made stronger by comparing on existing settings, so that it's possible to compare against existing numbers in papers. Some of the performance benefits of the proposed method could be explained simply by insufficient tuning of the baseline models.
- It would be good to see further ablation to test some of the claims made in the paper. For example, it would be good to check the performance impact of the parallel computation vs sequential computation.

[1] https://arxiv.org/pdf/2004.03990.pdf

**Questions:**

It seems like using all possible subsets can be limiting in terms of scalability. Can you think of more efficient alternatives that don't sacrifice too much in model capacity?

---

> ### Author Response · Authors · 2023-11-20
> **The difference between high-order permutation equivariance and high-dimensional permutation equivariance**
>
> Thank you for your thorough reviews and valuable feedback.
>
> The **high-order** permutation equivariance considers the high-order
> features of a single type of object. Assuming there are $K$ objects,
> their $N$th-order features among each other are denoted as
> ${{{\bf{\mathsf{X}}}}}\in{\mathbb R}^{K\times K\times \cdots\times K\times D\_X}$,
> where $K$ is repeated $N$ times in the dimension, so it can also be
> written as ${\bf{\mathsf{X}}}\in{\mathbb R}^{K^N\times D\_X}$ \[1\]. The
> function $f$ exhibits $N$th-order permutation equivariance if it
> satisfies $$\begin{aligned}
>     f(\sigma\circ{\bf{\mathsf{X}}})=\sigma\circ f({\bf{\mathsf{X}}}),\ (\sigma\circ{\bf{\mathsf{X}}})\_{[k\_1,...,k\_N,:]} = {\bf{\mathsf{X}}}\_{[\sigma^{-1}\circ k\_1,...,\sigma^{-1}\circ k\_N,:]},\ \sigma\in {\mathbb S}\_K.
>     \end{aligned}$$ It can be observed that the same permutation
> $\sigma$ is employed across all dimensions. **High-order** permutation
> equivariance is an extension of **one-dimensional** permutation
> equivariance \[1\]\[2\]. The relevant literature on high-order
> equivariance also includes references \[3\], \[4\], \[5\], and so on.
> The **multi-dimensional** equivariance considers features among multiple
> types of objects, i.e.,
> ${{{\bf{\mathsf{X}}}}}\in{\mathbb R}^{K\_1\times K\_2\times \cdots\times K\_N\times D\_X}$.
> A mapping $f$ is said to be $N$-dimensional equivariant if
> $$\begin{aligned}
>     f\left({\pi}^{\mathbb N}\circ{{{\bf{\mathsf{X}}}}}\right) = {\pi}^{\mathbb N}\circ f({{{\bf{\mathsf{X}}}}}),\ ({\pi}^{\mathbb N}\circ {\bf{\mathsf{X}}})\_{[k\_1,k\_2,...,k\_N,:]} = {{\bf{\mathsf{X}}}}\_{[\pi^{-1}\_{K\_1}(k\_1),\pi^{-1}\_{K\_2}(k\_2),...,\pi^{-1}\_{K\_N}(k\_N),:]},\ \forall {\pi}^{\mathbb N}\in {\mathbb S}^{\mathbb N}.
>     \end{aligned}$$ **multi-dimensional** permutation equivariance
> considers different permutations $\pi\_{K\_1},...,\pi\_{K\_N}$ across
> multiple dimensions, which is clearly distinct from high-order
> permutation equivariance.
>
> \[1\] Maron H, Ben-Hamu H, Shamir N, et al. Invariant and equivariant
> graph networks\[J\]. arXiv preprint arXiv:1812.09902, 2018.
>
> \[2\] Thiede E H, Hy T S, Kondor R. The general theory of permutation
> equivarant neural networks and higher order graph variational
> encoders\[J\]. arXiv preprint arXiv:2004.03990, 2020.
>
> \[3\] Keriven N, Peyré G. Universal invariant and equivariant graph
> neural networks\[J\]. Advances in Neural Information Processing Systems,
> 2019, 32.
>
> \[4\] Pan H, Kondor R. Permutation equivariant layers for higher order
> interactions\[C\]. International Conference on Artificial Intelligence
> and Statistics. PMLR, 2022: 5987-6001.
>
> \[5\] Kim J, Oh S, Hong S. Transformers generalize deepsets and can be
> extended to graphs & hypergraphs\[J\]. Advances in Neural Information
> Processing Systems, 2021, 34: 28016-28028.

---

> ### Author Response · Authors · 2023-11-20
> **The novelty of the proposed framework**
>
> In this work, **we propose a plug-and-play paradigm for the construction
> of arbitrary high-dimensional equivariant networks with well-designed
> one-dimensional equivariant networks**. Specifically, we first provide a
> definition of multi-dimensional equivariance based on algebraic
> definitions. Based on this, we further propose two methods to combine
> one-dimensional equivariant functions, which have been mathematically
> proven to satisfy the equivariant property in multiple dimensions.
> Moreover, we employed feature reuse and introduced global information at
> each layer to enhance performance while maintaining the
> multi-dimensional equivariance. Additionally, it has been proven that
> the constructed equivariant layer can degenerate into the
> multi-dimensional equivariant linear layer characterized by a complex
> mathematical expression. The substitutability of one-dimensional
> equivariant functions in the proposed framework implies its enhanced
> generality, effectively transforming the design problem of
> multi-dimensional equivariant networks into the selection of
> one-dimensional equivariant networks.

---

> ### Author Response · Authors · 2023-11-20
> **Comparison of experimental setting**
>
> The hyperparameters used in our experiments are listed in Appendix E.4
> of the revised paper. These hyperparameters of the baseline models have
> been fine-tuned, and we reported the best results for comparison. For
> example:
>
> -   Pooling functions: Without loss of generality, all the pooling
>     operations are set to be the mean function.
>
> -   Activation & normalization: To accelerate convergence and enhance
>     the stability of the training process, we applied the ReLU
>     activation and LN to $\mathtt{ETN2D}$ and $\mathtt{ETN3D}$.
>
> -   Parameter initialization: With the exception of networks
>     $\mathtt{SMPEN2D-TF}$ and $\mathtt{SMPEN3D-TF}$ in the task
>     of Section 4.3, which were initialized using the default method, we
>     employed Kaiming initialization for the weight parameters of the
>     linear layers in networks used for other tasks.
>
> -   Monte-Carlo simulation: Each experimental result is the average of
>     three valid experiments.

---

> ### Author Response · Authors · 2023-11-20
> **Ablation experiments for the parallel computation and sequential computation**
>
> The ablation experiments have been included in Appendix F.1 of the
> revised paper. Specifically, to validate the superiority of
> $g\_{\rm serial}$ (represented by $\mathtt{ParaN2D-TF}$) over
> $g\_{\rm para}$ (represented by $\mathtt{SerialN2D-TF}$), as well as
> the superiority of $g\_{\rm SMPE}$ over $g\_{\rm serial}$, we compared the
> performance of these three methods on the task in Section 4.2. In most scenarios, as observed from the table below, $g\_{\rm SMPE}$ outperforms $g\_{\rm serial}$, which, in turn, outperforms $g\_{\rm para}$.
>
> | **Test/Train Setting**           | $\\mathcal{A}$/$\\boldsymbol{\\mathcal{A}}$ | $\\mathcal{A}$/$\\mathcal{B}$ | $\\mathcal{A}$/$\\mathcal{C}$ | $\\mathcal{B}$/$\\mathcal{A}$ | $\\mathcal{B}$/$\\boldsymbol{\\mathcal{B}}$ | $\\mathcal{B}$/$\\mathcal{C}$ | $\\mathcal{C}$/$\\mathcal{A}$ | $\\mathcal{C}$/$\\mathcal{B}$ | $\\mathcal{C}$/$\\boldsymbol{\\mathcal{C}}$ |
> | :------------------------------: | :---------------------------------------------: | :-------------------------------: | :-------------------------------: | :-------------------------------: | :---------------------------------------------: | :-------------------------------: | :-------------------------------: | :-------------------------------: | :---------------------------------------------: |
> | $\\mathtt{ParaN2D\\!-\\!TF}$   | 20\.49                                          | 20\.39                            | 20\.33                            | 21\.72                            | 21\.87                                          | 21\.62                            | 21\.99                            | 21\.61                            | 21\.97                                          |
> | $\\mathtt{SerialN2D\\!-\\!TF}$ | 20\.75                                          | 21\.48                            | 19\.08                            | 21\.91                            | 22\.94                                          | 20\.31                            | 19\.26                            | 20\.18                            | 24\.33                                          |
> | $\\mathtt{SMPEN2D\\!-\\!TF}$   | **22\.21**                                      | 22\.70                            | 20\.69                            | 23\.78                            | **24\.71**                                      | 22\.30                            | 21\.80                            | 23\.17                            | **24\.44**                                      |

---

> ### Author Response · Authors · 2023-11-20
> **More efficient alternatives without sacrificing too much model capacity**
>
> First, we would like to illustrate the primary sources of complexity in
> the framework. The computational complexities and parameter quantities
> are shown as follows
>
> | Layer Type              | Complexity                                                                             | Parameters                             |
> |-------------------------|----------------------------------------------------------------------------------------|----------------------------------------|
> | $\mathtt{SMPEL-DS}$ | $\mathcal{O}\left((\Pi^{N}_{n=1}K_n)\cdot(2^N+N+1)\cdot D^2\right)$                    | $\mathcal{O}\left((2^N-1+N)D^2\right)$ |
> | $\mathtt{SMPEL-TF}$ | $\mathcal{O}\left((\Pi^{N}\_{n=1}K_n)\cdot[(2^N+N-1)D+\sum_{n=1}^{N}K_n]\cdot D\right)$ | $\mathcal{O}\left((2^N-1+N)D^2\right)$ |
> | $\mathtt{ETL}$          | $\mathcal{O}\left((\Pi^{N}_{n=1}K_n)\cdot2^N\cdot D^2\right)$                          | $\mathcal{O}(2^ND^2)$                  |
>
> The total number of input features is $\Pi\_{n=1}^NK\_n$. This implies
> that under the condition of a fixed input dimensions $N$, the
> computational complexity orders of $\mathtt{SMPEL-DS}$ and
> $\mathtt{SMPEL-TF}$ are lower than the second order of
> $\Pi\_{n=1}^NK\_n$, and the parameter quantity is also independent of
> $\Pi\_{n=1}^NK\_n$. Therefore, the main factor that limits the scalability
> of the framework is the exponential expansion of network size with the
> growth of dimension $N$. The number $2^N$ in this table arises from the
> algorithm's requirement to obtain pooled information for every subset in
> the power set of ${\mathbb N}\backslash\{\emptyset\}$ and incorporate it
> into the computation.
>
> The exploration of efficient alternatives that do not significantly
> compromise model capacity is an interesting topic. We are currently
> conducting further research about this problem. From a practical
> implementation perspective, the following method can achieve a trade-off
> between performance and complexity: Train the network for multiple
> times, with each training randomly selecting $H$ subsets from the
> $2^N-1$ subsets to include in the computation. Then, the best performing
> networks are retained. This approach allows for the identification of
> the most important $H$ combinations, effectively reducing the scale from
> $2^N-1$ to $H$.

---

### Official Review · Reviewer_9EiR · 2023-11-01

**Soundness:** 3 good
**Presentation:** 2 fair
**Contribution:** 2 fair
**Rating:** 6
**Confidence:** 3

**Summary:**

This paper introduces SMPE, a multi-dimensional permutation equivariant framework. The framework combines 1D equivariant functions applied to individual dimensions to preserve higher-dimensional equivariance. By incorporating feature reuse and cross-dimensional global information, SMPE facilitates interactions among objects in all dimensions, enhancing expressivity. Additionally, the framework can be extended to achieve multi-dimensional invariance using pooling operations. Experimental results show that networks constructed using the SMPE framework outperform other approaches and can operate on sets of varying sizes.

**Strengths:**

* It is valuable to preserve higher-dimensional equivariance.
* The theoretical analyses are provided.
* The experimental results show the method have significant improvements over the baselines.

**Weaknesses:**

1. One concern is about the evaluation. 1) The adopted tasks in this paper consider the dimensions no more than 3D. This may not reflect the effectiveness of the proposed method, since the paper claims its contribution in **high-dimensional** permutation equivariance. 2) The tasks seem simple that they only consider a linear mapping. It would be better to evaluate the method in real-world datasets, where the mapping from inputs to outputs could be complex. It would be better to show the method can be an effective module in neural networks for complex tasks.

2. Another concern is about the novelty. The idea seems straightforward to combine 1D equivariant functions applied to individual dimensions to preserve higher-dimensional equivariance.

**Questions:**

1. Can this method act a high-dimensional permutation equivariant module in other neural networks for more complex tasks?

2. Most graph neural networks are known as permutation equivariant, and most pooling methods are permutation invariant. What about the comparisons of this method with these works?

3. What is the distance metric for calculating MAE&MSE in the experiments?

4. What about the tasks with more dimensions, e.g., more than 3D?

---

> ### Author Response · Authors · 2023-11-20
> **The effectiveness of the proposed method on tasks with more than 3D**
>
> We sincerely thank you for your careful reviews and constructive comments.
>
> The proposed framework can be easily extended to higher dimensions, and
> we illustrate the framework of network construction by using a task of
> 4D mapping as an example.
>
> Consider the precoding design for cell-free multi-user multiple-input
> multiple-output (MU-MIMO) systems There is a CPU controlling $M$
> distributed APs to support $K$ UEs, where each AP is equipped with
> $N\_{T}$ antennas and each AP is equipped with $N\_{R}$ antennas. The
> channel between the $m$-th AP and the $k$-th UE is denoted as
> ${{\bf H}}\_{k,m}\in{\mathbb{C}}^{N\_{T}\times N\_{R}}$, and the
> corresponding precoding matrix is
> ${\bf W}\_{k,m}\in{\mathbb{C}}^{N\_{T}\times N\_{R}}$. By stacking the
> channel and precoding matrices belonging to all APs and UEs, we have
> ${{\bf{\mathsf{H}}}}\in{\mathbb{C}}^{M\times K\times N\_{T}\times N\_{R}}$
> and
> ${{\bf{\mathsf{W}}}}\in{\mathbb{C}}^{M\times K\times N\_{T}\times N\_{R}}$.
> Similar to the task in Section 4.3, we consider the optimization problem
> of maximizing the sum rate under transmit power constraints. It can be
> proved that the mapping from ${{\bf{\mathsf{H}}}}$ to certain optimal
> ${\bf{\mathsf{W}}}^{\star}=f({{\bf{\mathsf{H}}}})$ is 4D permutation
> equivariant.
>
> For this task, we concatenate the real parts and imaginary parts of
> ${{\bf{\mathsf{H}}}}$, and the shapes of the input and the target output
> are
> ${\widetilde {{\bf{\mathsf{H}}}}}\in{\mathbb R}^{M\times K\times N\_{T}\times N\_{R}\times 2}$
> and
> ${\widetilde {{\bf{\mathsf{W}}}}}\in{\mathbb R}^{M\times K\times N\_{T}\times N\_{R}\times 2}$,
> respectively. For this, the architecture of layer
> ${\rm SMPEL}:{\mathbb R}^{M\times K\times N\_{T}\times N\_{R} \times D}\to {\mathbb R}^{M\times K\times N\_{T}\times N\_{R}\times D}$
> can be represented as follows: $$\begin{aligned}
>         {\rm SMPEL}({\bf{\mathsf{X}}}) = {\rm FC}\_1(\sigma({\rm LN}({\bf{\mathsf{M}}}))),\\
>         {\bf{\mathsf{M}}} = {\rm FC}\_2([{\bf{\mathsf{O}}}\_1, {\bf{\mathsf{O}}}\_2, {\bf{\mathsf{O}}}\_3, {\bf{\mathsf{O}}}\_4, {\bar {\bf{\mathsf{X}}}}\_{{\mathbb P}\_1}, ..., {\bar {\bf{\mathsf{X}}}}\_{{\mathbb P}\_{15}}]),\\
>         {\bf{\mathsf{O}}}\_0 = {\bf{\mathsf{X}}},\ {\bf{\mathsf{O}}}\_{n} = {\rm PE1D}\_{n}({\bf{\mathsf{O}}}\_{n-1}),\ n\in\{1, 2, 3, 4\},
>         \end{aligned}$$ where
> ${\rm FC}\_1:{\mathbb R}^{D}\to{\mathbb R}^{D}$ and
> ${\rm FC}\_2:{\mathbb R}^{(4+15)D}\to{\mathbb R}^{D}$ represent the
> linear layer applied at the last dimension.
> ${\rm PE1D}\_1:{\mathbb R}^{M\times D}\to {\mathbb R}^{M\times D}$,
> ${\rm PE1D}\_2:{\mathbb R}^{K\times D}\to {\mathbb R}^{K\times D}$,
> ${\rm PE1D}\_3:{\mathbb R}^{N\_{T}\times D}\to {\mathbb R}^{N\_{T}\times D}$,
> and
> ${\rm PE1D}\_4:{\mathbb R}^{N\_{R}\times D}\to {\mathbb R}^{N\_{R}\times D}$
> denotes the 1D equivariant layer operating at the 1,2,3, and 4-th
> dimensions, respectively.
>
> Then, the four-dimensional equivariant neural network used to model $f$
> can be constructed as follows $$\begin{aligned}
>         {\widetilde {{\bf{\mathsf{W}}}}} = \sigma\_{\rm out}({\rm FC}\_{\rm out}({\rm SMPEN}({\rm FC}\_{\rm in}({\widetilde {{\bf{\mathsf{H}}}}})))),
>         \end{aligned}$$ where
> ${\rm FC}\_{\rm in}:{\mathbb R}^{2}\to {\mathbb R}^{D}$ and
> ${\rm FC}\_{\rm out}:{\mathbb R}^{D}\to {\mathbb R}^{2}$ are used to is
> used to change the feature dimensions. $\sigma\_{\rm out}(\cdot)$ is
> designed to ensure that ${\widetilde {{\bf{\mathsf{W}}}}}$ satisfies the
> power constraints. At this point, the construction of the
> four-dimensional permutation-equivariant network from
> ${\widetilde {{\bf{\mathsf{H}}}}}$ to ${\widetilde {{\bf{\mathsf{W}}}}}$
> is completed. Following this approach, the framework can be easily
> extended to higher dimensions.

---

> ### Author Response · Authors · 2023-11-20
> **The nonlinearity and complexity of the mappings considered in the tasks**
>
> The involved mappings are relatively complex non-linear mappings. A
> linear mapping between two vector spaces needs to satisfy additivity,
> which means $f(X+Y)=f(X)+f(Y)$. Clearly, the pseudo-inverse and the
> optimal solution for downlink transmission design do not satisfy this
> property. Specifically, for the transmission design task in Sections 4.2
> and 4.3, finding the optimal results requires solving Problem (26) and
> Problem (28), respectively. These two problems are typical NP-hard
> problems and are difficult to solve \[1\]. The typical solving
> algorithms usually involve iterative operations on high-dimensional
> matrices and require multiple initializations to select the optimal
> results \[2\], which is very time consuming and can not guarantee
> optimality.
>
> \[1\] Zhao X, Lu S, Shi Q, et al. Rethinking WMMSE: Can its complexity
> scale linearly with the number of BS antennas?\[J\]. IEEE Transactions
> on Signal Processing, 2023, 71: 433-446.
>
> \[2\] Christensen S S, Agarwal R, De Carvalho E, et al. Weighted
> sum-rate maximization using weighted MMSE for MIMO-BC beamforming
> design\[J\]. IEEE Transactions on Wireless Communications, 2008, 7(12):
> 4792-4799.

---

> ### Author Response · Authors · 2023-11-20
> **Evaluation of the proposed method on real-world datasets**
>
> In order to enhance the practicality and complexity of the task, we
> incorporated real-world datasets and conducted experiments on the
> transmission design task discussed in Section 4.3. We replaced the
> original results in Table 3 with the new results and move the original
> Table 3 to the appendix as supplementary results.
>
> The new dataset is provided by QuaDRiGa \[1\]. QuaDRiGa, short for QUAsi
> Deterministic RadIo channel GenerAtor, is used for generating realistic
> radio channel impulse responses for system-level simulations of mobile
> radio networks. These simulations are used to determine the performance
> of new digital-radio technologies in order to provide an objective
> indicator for the standardization process in bodies like 3rd Generation
> Partnership Project (3GPP). Furthermore, the channel generator is
> continuously updated with the evolution of wireless communication
> standards \[2\]. For more details, please refer to the official website
> of QuaDRiGa: https://quadriga-channel-model.de.
>
> The dataset configuration we used is as follows
>
> |                              |                                            |                    |        |   |
> |:----------------------------:|:------------------------------------------:|:------------------:|:------:|:---:|
> | Scenario                     | 3GPP_38.901_UMa_NLOS                       | Center Frequency   | 3.5GHz |   |
> | Transmit Antenna             | 3gpp-3d                                    | Receive Antenna    | omni   |   |
> | AP Height                    | 25m                                        | User Height        | 1.5m   |   |
> | Transmitter Antenna Downtilt | 10                                         | Subcarrier Spacing | 30kHz  |   |
> | AP/User Distribution         | Uniformly distributed within a 500m radius |                    |        |   |
> |                              |                                            |                    |        |   |
>
>
> where the scenario '3GPP\\_38.901\\_UMa\\_NLOS ' refers to a specific
> configuration defined by the 3GPP in technical specification 38.901
> \[3\]. This scenario is designed to model wireless communication in
> urban macro environments where non-line-of-sight (NLOS) conditions are
> prevalent. In this scenario, QuaDRiGa simulates a multi-path channel
> model that includes the effects of buildings, structures, and other
> obstacles typically found in urban macro environments. The model
> considers the propagation of radio waves in NLOS conditions, accounting
> for reflections, diffractions, and scattering from surrounding objects.
>
> The dataset from the generator closely approximates real-world
> scenarios, ensuring the practical value of experimental results.
>
> \[1\] Jaeckel S, Raschkowski L, Börner K, et al. QuaDRiGa: A 3-D
> multi-cell channel model with time evolution for enabling virtual field
> trials\[J\]. IEEE transactions on antennas and propagation, 2014, 62(6):
> 3242-3256.
>
> \[2\] Wang C X, Bian J, Sun J, et al. A survey of 5G channel
> measurements and models\[J\]. IEEE Communications Surveys & Tutorials,
> 2018, 20(4): 3142-3168.
>
> \[3\] 3GPP TR 38.901 v16.1.0, "Study on channel model for frequencies
> from 0.5 to 100 GHz,\" Tech. Rep., 2019.

---

> ### Author Response · Authors · 2023-11-20
> **The novelty of the proposed framework that seems straightforward to combine 1D equivariant functions**
>
> There are various ways to apply one-dimensional equivariant functions to
> multiple dimensions, but not all of them can maintain the equivariant
> property in multiple dimensions. For example, one can flatten the
> equivariant dimensions of the multi-dimensional data and apply
> one-dimensional equivariant functions. Alternatively, one can fix one
> equivariant dimension and flatten the remaining dimensions along with
> the feature dimension to create a new feature dimension. However, these
> approaches do not satisfy the multi-dimensional equivariance.
>
> In our work, **we propose a plug-and-play paradigm for the construction
> of arbitrary high-dimensional equivariant networks with well-designed
> one-dimensional equivariant networks**. Unlike the examples mentioned,
> these proposed combination methods have been mathematically proven to
> satisfy the equivariant property in multiple dimensions. Moreover, we
> employed feature reuse and introduced global information at each layer
> to enhance performance while maintaining the multi-dimensional
> equivariance. Additionally, it has been proven that the constructed
> equivariant layers can degenerate into the multi-dimensional equivariant
> linear layers described in \[1\], which is characterized by a complex
> mathematical expression. In summary, while our proposed method has a
> straightforward expression, its foundation and underlying analysis are
> not straightforward. The concise expression reduces the cost of further
> exploitation and analysis, reflecting the principle we advocate.
>
> \[1\] Hartford J, Graham D, Leyton-Brown K, et al. Deep models of
> interactions across sets\[C\]. International Conference on Machine
> Learning. PMLR, 2018: 1909-1918.

---

> ### Author Response · Authors · 2023-11-20
> **Comparison of the proposed method with equivariant graph neural networks and invariant pooling methods**
>
> About graph neural networks: Graph neural networks are designed to solve
> graph-related problems, which often consider modeling of nodes and edges
> \[1\]. Due to the unordered nature of nodes, graph neural networks
> possess permutation equivariance. It is worth noting that most graph
> neural networks can be viewed as **one-dimensional** equivariant
> networks, as evidenced by references \[2\] and \[3\]. As our work
> primarily focuses on the construction of multi-dimensional equivariant
> networks, our approach is more general than graph neural networks that
> target **one-dimensional** equivariant network design.
>
> About pooling methods: The pooling function is utilized to aggregate
> multiple features, and this characteristic can be employed to construct
> permutation-invariant networks. In Sections 3.2 and 3.3 of the paper,
> pooling functions are respectively introduced to acquire global
> information and provide invariance. Most pooling functions can be
> applied within the proposed framework, as the multi-dimensional
> equivariance maintained by our framework does not depend on the
> selection of pooling function. Therefore, designing pooling functions is
> beyond the scope of this paper.
>
> \[1\] Wu Z, Pan S, Chen F, et al. A comprehensive survey on graph neural
> networks\[J\]. IEEE transactions on neural networks and learning
> systems, 2020, 32(1): 4-24.
>
> \[2\] Vignac C, Loukas A, Frossard P. Building powerful and equivariant
> graph neural networks with structural message-passing\[J\]. Advances in
> neural information processing systems, 2020, 33: 14143-14155.
>
> \[3\] Velickovic P, Cucurull G, Casanova A, et al. Graph attention
> networks\[J\]. stat, 2017, 1050(20): 10-48550.

---

> ### Author Response · Authors · 2023-11-20
> **The distance metric for calculating MAE&MSE in the experiments**
>
> The computation methods of these two distance metrics are as follows
> $$\begin{aligned}
>     {\rm MAE} = (\|\Re({{\bf X}\_1})-\Re({\bf X}\_2)\|\_1+\|\Im({\bf X}\_1)-\Im({\bf X}\_2)\|\_1)/KN,\\
>     {\rm MSE} = (\|\Re({\bf X}\_1)-\Re({\bf X}\_2)\|^2\_F+\|\Im({\bf X}\_1)-\Im({\bf X}\_2)\|^2\_F)/KN,
>     \end{aligned}$$ where ${\bf X}\_1\in\mathbb{C}^{N\times K}$ and
> ${\bf X}\_2\in\mathbb{C}^{N\times K}$ are two matrices that need to be
> measured for their distance. It is worth noting that the upper and lower
> parts of Table I in the paper correspond to two separate experiments
> conducted for training networks with respect to these two metrics.
>
> These details are provided in Appendix E.1.

---

### Official Review · Reviewer_4oi5 · 2023-11-01

**Soundness:** 3 good
**Presentation:** 3 good
**Contribution:** 3 good
**Rating:** 6
**Confidence:** 2

**Summary:**

This paper proposes a novel serial multi-dimensional permutation equivariance framework called SMPE by serially composing multiple one-dimensional equivariant layers and incorporating dense connections for feature reuse to enable multi-dimensional interactions.

**Strengths:**

1. The proposed SMPEL seems to be reasonable.

2. The experimental results demonstrate the effectiveness of the proposed SMPE.

**Weaknesses:**

1. The pooling operation for multi-dimensional permutation invariance is not clear. Which type of pooling do you use in the proposed method? How does the pooling layer help with the multi-dimensional permutation invariance?

2. Some experimental setups are not clear. For instance, in Section 4.2, the authors do not explicitly mention what evaluation metric is used in Table 2. It's unclear which type of pooling is used in the proposed method.

**Questions:**

1. How does $\bar{X}_\mathbb{P}$ with the weight $\mathcal{w}^{GI}_P$ to preserve the global information? How do you learn the weights $\mathcal{w}^{GI}_P$ and $\mathcal{w}^{PE}_P$? Can you elaborate on this?

2. The pooling operation for multi-dimensional permutation invariance is not clear. Which type of pooling do you use in the proposed method? How does the pooling layer help with the multi-dimensional permutation invariance?

3. In the experiment, the authors provide various variants of SMPEN. For instance, in Table 3, SMPEN2D-TF sometimes outperforms SMPEN3D-TF, while SMPEN3D-TF achieve the best performance in different training settings. Given a task, how to determine which type of methods should be chosen to achieve the best performance?

---

> ### Author Response · Authors · 2023-11-20
> **The type of pooling function used for invariance**
>
> We sincerely appreciate your thorough reviews and insightful comments.
>
> In this work, pooling functions are used in two instances: Firstly, in
> Section 3.2, a pooling function is utilized to extract global
> information from the input of each layer. Secondly, in Section 3.3,
> pooling functions are introduced across multiple dimensions to transform
> multi-dimensional equivariant functions into multi-dimensional invariant
> functions. Without loss of generality, both of the two types are set to
> be the mean function in the experiments, since the design of the pooling
> function is not the focus of this work.
>
> We have supplemented this information in Section 4 and Appendex E.1.

---

> ### Author Response · Authors · 2023-11-20
> **The pooling layer's contribution to multi-dimensional invariance**
>
> Reference \[2\] demonstrates that a general invariant function can be
> decomposed into
> $f({\bf x})=\rho\left({\rm sum}(\phi(x\_1),...,\phi(x\_K))\right)$, where
> $\phi$ and $\rho$ are the suitable transformations. More generally,
> reference \[1\] reformulates this as
> $f({\bf x})=\rho\left({\rm pool}(\phi(x\_1),...,\phi(x\_K))\right)$.
> Furthermore, it was mentioned in \[1\] that if the transformation $\phi$
> appied at ${\bf x}$ is a stack of permutation equivariant layer, the
> model remains permutation invariant, and it is easy to prove that
> $f({\bf x})={\rm pool}({\rm PEL}({\bf x}))$ is a permutation invariant
> function, where ${\rm PEL}(\cdot)$ is the equivariant layer. Based on
> this, it can be demonstrated that a multi-dimensional equivariant
> function can be transformed into a multi-dimensional invariant function
> by pooling the output across multiple dimensions. Thus, we transform
> SMPEN into SMPIN in Section 3.3, expressed as
> ${\rm SMPIN}({\bf{\mathsf{X}}}) = {\rm Pool}\_{{\mathbb N}}\left({\rm SMPEN}({\bf{\mathsf{X}}})\right)$,
> where ${\rm Pool}\_{{\mathbb N}}$ represents the pooling function
> operating on dimensions in ${\mathbb N}$.
>
> \[1\] Lee J, Lee Y, Kim J, et al. Set transformer: A framework for
> attention-based permutation-invariant neural networks\[C\].
> International conference on machine learning. PMLR, 2019: 3744-3753.
>
> \[2\] Zaheer M, Kottur S, Ravanbakhsh S, et al. Deep sets\[J\]. Advances
> in neural information processing systems, 2017, 30.

---

> ### Author Response · Authors · 2023-11-20
> **The experimental setup, including the evaluation criteria in Table 2 and the used pooling layer**
>
> Due to the limitations of the main text, we have added Appendix E titled
> \"Experimental Setup\" in the revised version, which includes the
> specific mathematical expressions for the evaluation metric of each
> experiment and the selection of used pooling operations. For example,
> the evaluation metric ${\bar R}\_{\rm M}$ in Table 2 is defined as the
> average of $R\_{\rm M}$ across seven SNR values
> $\{0, 5, 10, 15, 20, 25, 30\}$, where the expression of $R\_{\rm M}$ is
> given by $$\begin{aligned}
>     R\_{\rm M}=\sum\_{k=1}^{K}\log\_2\left(1+\frac{|{\bf h}\_k^H{{\bf w}}\_k|^2}{\sigma^2+\sum\_{i\neq k}|{\bf h}^H\_k{\bf w}\_i|}\right),
>     \end{aligned}$$
> where the definitions of symbols can be found in
> Section D.2.

---

> ### Author Response · Authors · 2023-11-20
> **The approach for ${\bar{\bf{\mathsf{X}}}}\_{{\mathbb P}}$ and weights to preserve global information**
>
> ${\bar{\bf{\mathsf{X}}}}\_{{\mathbb P}}$ is obtained from pooling
> functions, which means it is aggregated from the information of each
> object. Its invariance to objects implies that it contains the global
> information of all object features. In each layer of the network,
> ${\bar{\bf{\mathsf{X}}}}\_{{\mathbb P}}$ is connected to the output after
> being weighted by learnable parameters $w^{\rm GI}$. This means that
> after passing through this layer, the global information of the input is
> still preserved in the output of this layer. Similarly, by stacking
> layers, the global information of the input is preserved in the network.

---

> ### Author Response · Authors · 2023-11-20
> **The learning approach for weights $w^{GI}$ and $w^{PE}$**
>
> The weighted sum of the processed features and the global information is
> performed through equations (5) in the paper. The processing of the
> linear layer ${\rm FC}\_2$ can be rewritten as $$\begin{aligned}
>     {\rm FC}\_2([{\bf{\mathsf{O}}}\_1, ..., {\bf{\mathsf{O}}}\_N, {\bar{\bf{\mathsf{X}}}}\_{{\mathbb P}\_1}, ..., {\bar{\bf{\mathsf{X}}}}\_{{\mathbb P}\_{2^N-1}}])=\sum\_{n=1}^{N}{\bf{\mathsf{O}}}\_n\times\_{(N+1)}{\bf W}^{\rm PE}\_n + \sum\_{i=1}^{2^N-1}{\bar{\bf{\mathsf{X}}}}\_{{\mathbb P}\_i}\times\_{(N+1)}{\bf W}^{\rm GI}\_i +\_{(N+1)} {\bf b},
>     \end{aligned}$$
>
> Where ${\bf W}^{\rm PE}\_n\in{\mathbb R}^{D\times D}$ and
> ${\bf W}^{\rm GI}\_i\in{\mathbb R}^{D\times D}$ are the matrix weights
> representing $w^{\rm PE}\_n$ and $w^{\rm GI}\_{{\mathbb P}\_i}$, and
> ${\bf b}\in{\mathbb R}^{D}$ is the bias of this layer. The operation
> '$\times\_{(N+1)}$' denotes that the multiplication acts on the
> $(N+1)$-th dimension of the tensor (i.e., the last dimension) \[1\].
> Similarly, the bias is added along the last dimension, which does not
> affect the equivariance. By training this linear layer ${\rm FC}\_2$, the
> weights $w^{\rm PE}\_n$ and $w^{\rm GI}\_{{\mathbb P}\_i}$ can be learned.
>
> \[1\] Kolda, Tamara G., and Brett W. Bader. Tensor decompositions and
> applications. SIAM review 51(3), 455-500, 2009.

---

> ### Author Response · Authors · 2023-11-20
> **Selection of the network with the best performance for a given task**
>
> This is an interesting topic that is currently being considered.
> Although we are unable to provide rigorous mathematical conclusions at
> this moment, we can analyze the following points from an experimental
> perspective and offer practical recommendations.
>
> 1\. The impact of the number of objects $K\_1,...,K\_N$ on performance:\
> We consider the performance of a trained network from two perspectives:
> the testing performance in the training configuration and in other
> configurations. During testing in the training configuration, the number
> of of objects has a negligible impact on the network performance.
> However, when tested in other configurations, if the number of a certain
> type of objects in the training configuration is small, the change of
> this number will have a significant impact on the network performance.
> For example, in Table 3, the performance of SMPEN3D-TF trained under
> configuration ${\mathcal{B}}$ shows poor performance when tested under
> configuration ${\mathcal{A}}$. Conversely, if the number of such objects
> is large in the training configuration, minor variations in number have
> a negligible impact on the performance. For instance, in Table 2,
> SMPEN2D-TF trained under configuration ${\mathcal{B}}$ exhibits
> excellent performance when tested under configuration ${\mathcal{C}}$.
> Intuitively, when the number difference is small, the modeled
> equivariant mappings are similar, while the distinctions in the modeled
> mappings become more significant as the number difference grows large.
>
> 2\. The impact of the relationship between the number of object types $N$
> and the network dimensionality on performance:\
> When the dimension of the equivariant network is equal to the number of
> object types in the input, the input matches the network. With adequate
> training, the network can achieve good performance under the training
> configuration. When the dimension of the equivariant network is smaller
> than the number of object types in the input, multiple types of objects
> are reshaped into a single dimension for input. Due to equivariance, the
> network loses its ability to distinguish the merged types, resulting in
> a decrease in performance. For example, in Table 3, SMPEN3D-TF trained
> under configuration ${\mathcal{A}}$ performs better than SMPEN2D-TF when
> tested under configuration ${\mathcal{A}}$. However, merging multiple
> types of objects into a single type will increase the number of the new
> type of objects. According to the analysis of the first point, this
> operation may lead to better performance of the network when tested
> under other configurations. For instance, in Table 3, SMPEN2D-TF trained
> under configuration ${\mathcal{B}}$ performs better than SMPEN3D-TF when
> tested under configuration ${\mathcal{A}}$.
>
> In conclusion, we provide the following recommendations for network
> selection.
>
> -   If the number of each type of objectis is relatively large, choose
>     an equivariant network with the same dimension as the number of
>     types. For example, for the task in Table 2, we choose SMPEN2D-TF
>     instead of SetTrans1D.
>
> -   If the number of each type of objectis is relatively small, to
>     achieve better testing performance under the training configuration,
>     it is still advisable to choose an equivariant network with the same
>     dimension as $N$. To achieve better testing performance under other
>     configurations, selecting a network with lower equivariant dimension
>     can be considered. For instance, for the task in Table 3, SMPEN3D-TF
>     performs better under the training configuration, while SMPEN2D-TF
>     performs better under other configurations.
>
> Finally, many thanks for your careful reviews and constructive comments.

---

> > ### Comment · Reviewer_4oi5 · 2023-11-22
> > **Response to Authors**
> >
> > Thanks for the detailed clarification and I would like to raise my score to 6.

---

### Author Response · Authors · 2023-11-20
**Summary of changes**

We sincerely appreciate all the constructive comments. The main changes we have made are as follows:

1. We have added Appendix E (page 17-18) titled "Experimental Setup", which includes the specific mathematical expressions for the evaluation metric, the selection of pooling functions, other network hyperparameters, etc.

2. To enhance the practicality and complexity of the task, we incorporated real-world datasets and conducted experiments on the transmission design task discussed in Section 4.3 (page 9). The introduction to the datasets can be found in Appendix E.3 (page 17).

3. We have added ablation experiments about the networks constructed in a sequential or parallel manner in Appendix F.1 (page 18).

4. The running time comparison for networks across different tasks have been added to Appendix F.2 (page 18). This result demonstrates that our proposed methods have comparable runtimes to other methods.

5. The discussion on higher-order permutation equivariant networks has been added to Section 2.1, specifically in the ``Linear layers'' part (page 3).

6. The discussion on permutation equivariant graph neural networks for solving graph classification tasks has been added to Section 2.1, specifically in the ``Applications'' part (page 4).

---

### Meta-Review · Area_Chair_4moi · 2023-12-18

**Metareview:**

This paper proposes a serial multi-dimensional permutation equivariance framework for designing multi-dimensional equivariant mappings. While the reviewers found the results of the paper solid and useful, they also had some concerns on the novelty of the techniques and the scope of the numerical validation. Given all their scores are at the borderline and the need to evaluate more thoroughly the claims/changes made by the authors during the rebuttal process, I regrettably cannot recommend acceptance of the paper at this point.

**Justification For Why Not Higher Score:**

There are concerns on the novelty of the techniques and the scope of the numerical validation.

**Justification For Why Not Lower Score:**

N/A

---

### Decision · Program_Chairs · 2024-01-16

Reject